# Blood-brain barrier-restricted translocation of *Toxoplasma gondii* from cortical capillaries

Gabriela C Olivera, Emily C Ross, Christiane Peuckert, Antonio Barragan*

Department of Molecular Biosciences, The Wenner-Gren Institute, Stockholm University, Stockholm, Sweden

**Abstract** The cellular barriers of the central nervous system proficiently protect the brain parenchyma from infectious insults. Yet, the single-celled parasite *Toxoplasma gondii* commonly causes latent cerebral infection in humans and other vertebrates. Here, we addressed the role of the cerebral vasculature in the passage of *T. gondii* to the brain parenchyma. Shortly after inoculation in mice, parasites mainly localized to cortical capillaries, in preference over post-capillary venules, cortical arterioles or meningeal and choroidal vessels. Early invasion to the parenchyma (days 1-5) occurred in absence of a measurable increase in blood-brain barrier (BBB) permeability, perivascular leukocyte cuffs or hemorrhage. However, sparse focalized permeability elevations were detected adjacently to replicative parasite foci. Further, *T. gondii* triggered inflammatory responses in cortical microvessels and endothelium. Pro- and anti-inflammatory treatments of mice with LPS and hydrocortisone, respectively, impacted BBB permeability and parasite loads in the brain parenchyma. Finally, pharmacological inhibition or Cre/*lox*P conditional knockout of endothelial focal adhesion kinase (FAK), a BBB intercellular junction regulator, facilitated parasite translocation to the brain parenchyma. The data reveal that the initial passage of *T. gondii* to the central nervous system occurs principally across cortical capillaries. The integrity of the microvascular BBB restricts parasite transit, which conversely is exacerbated by the inflammatory response.

*For correspondence:
antonio.barragan@su.se

Competing interest: The authors declare that no competing interests exist.

## Editor's evaluation

There is significant interest in how pathogens like Toxoplasma infect the brain. The present study carefully details early steps in this process by inoculating mice intravenously and monitoring their distribution within brain tissues over the subsequent days. The analysis presented will be a valuable reference for further investigation of this process. The authors perform several perturbations that involve the permeability of the blood-brain barrier as a key determinant of parasite entry into the brain parenchyma. Taken together the study highlights the critical role of the blood-brain barrier in restricting access of Toxoplasma to the brain.

## Introduction

The vertebrate central nervous system (CNS) is protected by restrictive cellular barriers that maintain homeostasis and regulate passage of molecules and cells to the brain parenchyma (*Abbott et al., 2006*). Three main cellular barrier systems protect neurons from blood-borne external insults, such as infection: the blood-brain barrier (BBB), the blood-cerebrospinal fluid barriers and the meningeal barriers (*Coureuil et al., 2017*). Of these, the BBB exerts the greatest immediate impact on the cerebral microenvironment.

The BBB is anatomically localized to the cerebral microvasculature, which is constituted by capillaries with a luminal diameter (Ø) of <10 µm, arterioles and venules (~10–100 µm Ø) (*Wilhelm et al., 2016*; *Gould et al., 2017*). The microvascular lumen is covered by highly specialized endothelial cells characterized by low permeability, low pinocytic activity and high transcellular electrical resistance (TCER) (*Stamatovic et al., 2008*). Basal lamina, astrocyte 'end-feet' and pericytes also contribute to restrict permeability (*Daneman and Prat, 2015*; *Armulik et al., 2010*). Intercellular tight junction (TJ) proteins seal the spaces between adjacent endothelial cells and separate the apical from the basolateral side. This restricts the paracellular flux of hydrophilic molecules and regulates the migration of cells across the endothelial barrier (*Alon and van Buul, 2017*). Focal adhesion kinase (FAK), also named protein tyrosine kinase 2 (PTK2), regulates signaling pathways involved in cell-cell adhesion and focal adhesion complexes in endothelial cells (*Lee et al., 2010*; *Siu et al., 2009*). The turnover of FAK has been shown to regulate barrier function, and dysregulation of FAK impacts the integrity of cellular barriers (*Ivey et al., 2009*) by increasing paracellular permeability, which can imply disruption of the barrier function (*Ma et al., 2013*).

Inflammation can cause impairment of the blood-CNS barrier functions (*Abbott et al., 2006*). Immune activity by resident cells and infiltrating leukocytes are crucial to eradicate pathogens that succeed in translocating across the restrictive CNS barriers (*Coureuil et al., 2017*), but this response can also alter neuronal function (*Dando et al., 2014*). Therefore, striking a balance between controlling infection and limiting excessive inflammation is crucial (*Klein and Hunter, 2017*).

The obligate intracellular parasite *Toxoplasma gondii* infects a broad range of warm-blooded vertebrates, with an estimated 1/3 of the global human population being chronically infected (*Pappas et al., 2009*). From the point of entry in the intestinal tract, *T. gondii* rapidly achieves systemic dissemination and establishes latent infection in the CNS (*Schlüter and Barragan, 2019*). While chronic infection is generally considered asymptomatic, reactivated or acute infection can lead to life-threatening encephalitis in immune-compromised individuals and to neurological disorders in neonates (*Montoya and Liesenfeld, 2004*). *T. gondii* tachyzoites use gliding motility to actively invade cells where they replicate (*Dobrowolski and Sibley, 1996*). Gliding motility also facilitates transmigration of tachyzoites across the endothelium in vitro (*Ross et al., 2019*; *Barragan et al., 2005*) and provides an effective mechanism of propulsion in the microenvironment inside tissues (*Barragan and Sibley, 2002*). In their journey to form chronic cysts, the tachyzoites cross the cellular brain barriers to establish latent infection in the CNS (*Feustel et al., 2012*).

In recent years, work in the field has focused on parasite dissemination pathways and mechanisms of passage to the brain (*Konradt et al., 2016*; *Drewry et al., 2019*; *Kanatani et al., 2017*; *Bhandage et al., 2020*; *Schneider et al., 2019*; *Sangaré et al., 2019*). Yet, the roles played by the blood-CNS barriers have remained unresolved. Here, we address the implication of the BBB in the early passage of *T. gondii* tachyzoites to the CNS. Specifically, we assess the loci of passage of *T. gondii* across the cerebral cellular barriers and the impacts of BBB integrity, dysregulation and inflammation on the access of tachyzoites to the parenchyma. We identify a role for the TJ regulator FAK in the passage of *T. gondii* to the brain parenchyma.

## Results

### *T. gondii* tachyzoites localize to parenchymal capillaries in the cerebral microcirculation

*T. gondii* reaches the peripheral blood circulation shortly after infection (*Derouin and Garin, 1991*) and localization within the cerebral vasculature precedes its penetration to the brain parenchyma in experimental infections (*Konradt et al., 2016*). To study the distribution of *T. gondii* in the brain vasculature, we inoculated mice in the tail vein with GFP-expressing *T. gondii* tachyzoites and a fluorescent intraluminal vascular tracer (Evans blue, EB). Interestingly, immuno-histochemical analyses of brain tissue revealed that *T. gondii* tachyzoite GFP+ signal preferentially localized to microvessels with an intraluminal diameter less than 10 µm ( < 10 µm Ø; 91.32% ± 5.97) (*Figure 1a and b*). Furthermore, *T. gondii* tachyzoite signal preferentially localized to microvessels with low relative expression of the endothelial basement membrane marker α5 laminin (91.33% ± 8.08) (*Figure 1c and d*, *Figure 1— figure supplement 1*) and with high expression of P-glycoprotein relative to α5 laminin expression (67.60% ± 8.83) (*Figure 1e and f*). The accumulation of *T. gondii* tachyzoites in microvasculature

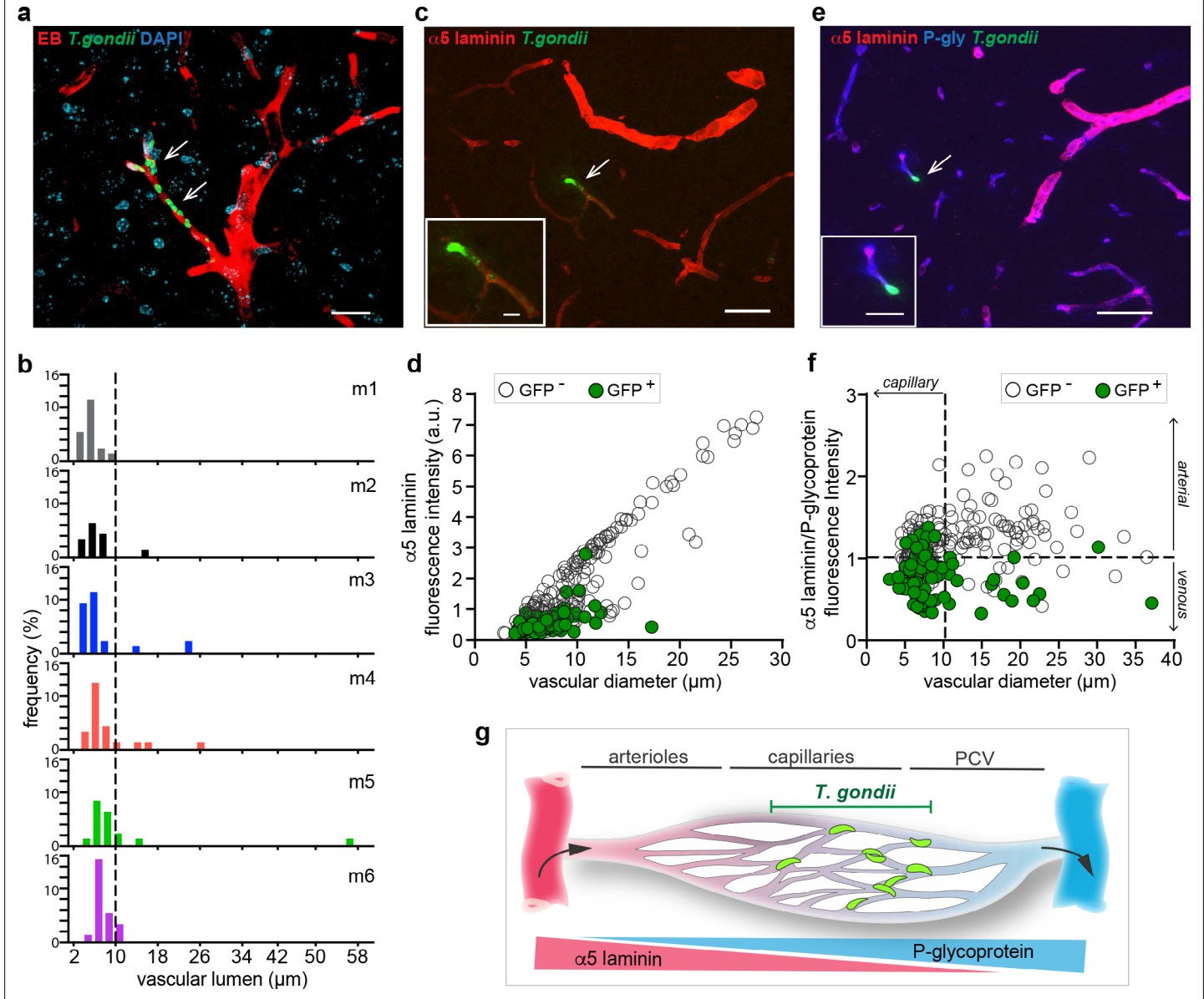

**Figure 1.** Localization of *T. gondii* tachyzoites in the cortical vasculature. C57BL/6 mice were challenged i.v. with 3–10 × 10⁶ *T. gondii* GFP-expressing tachyzoites (RH) and brains were extracted 72 hr post-inoculation. (**a**) Representative confocal micrograph of cortical section shows intraluminal vascular staining (EB) and nuclear staining (DAPI). Arrowheads indicate localization of tachyzoites. Scale bar, 20 µm. (**b**) Graphs show the relative frequency (%) of blood vessels (EB⁺) with associated *T. gondii* tachyzoites (GFP⁺) related to the luminal diameter (µm). Data is from three tissue sections per mouse from 6 mice (m1-6) ranging between 14 and 25 GFP⁺ vessels per mouse. (**c**) Immunofluorescence staining of cortical section for α5 laminin. Arrowhead and inset show *T. gondii* (GFP⁺) in vicinity of a vessel with relatively low α5 laminin signal. Scale bars, 50 µm, inset 10 µm. (**d**) Quantification of the relative fluorescence intensity (arbitrary units, a.u.) of α5 laminin signal in GFP⁻ and GFP⁺ vasculature, related to the blood vessel diameter (µm). Data is from a total of 284 vessels in three sections per mouse (n = 3 mice). ANCOVA, F (1, 281) = 36.931, p < 0.0001. (**e**) Representative immunofluorescence micrograph of cortical section stained for α5 laminin and P-glycoprotein (P-gly). Arrowhead and inset show *T. gondii* (GFP⁺) in the vicinity of a vessel. Scale bars, 50 µm, inset 10 µm. (**f**) Quantification of the relative fluorescence intensity (a.u.) ratio α5 laminin / P-glycoprotein, related to the blood vessel diameter (µm). Data is from a total of 275 vessels in three sections per mouse (n = 3 mice). ANCOVA, F (1, 272) = 80.255, p< 0.0001. (**g**) Schematic representation of the cortical microvasculature illustrates the preferential localization of *T. gondii* to capillaries and, to a lesser extent, post-capillary venules (PCV) in the brain parenchyma. Triangles indicate the relative expression of the vascular markers α5 laminin and P-glycoprotein in arterioles, capillaries, and PCV.

The online version of this article includes the following figure supplement(s) for figure 1:

**Figure supplement 1.** Representative epifluorescence micrographs of α5 laminin and α4 laminin co-stainings in isolated brain microvessels.

with <10 μm Ø and low relative α5 laminin/P-glycoprotein expression is consistent with preferential localization to parenchymal capillaries. *T. gondii* also localized to post-capillary venules ( > 10 μm Ø, low relative α5 laminin/P glycoprotein ratio) at a lower relative frequency compared with capillaries, while localization to penetrating arterioles, pre-capillary arterioles and penetrating venules was absent or scarce (*Figure 1f and g*).

## Distribution of parasite loads in brain parenchyma, choroid plexus (CP), meninges, and peripheral organs

Because the vasculature of peripheral organs and of the blood-CNS barriers exhibit very different restrictiveness to passage of cells (*Stamatovic et al., 2008*), we assessed the parasite loads at organ level and sub-organ level in brain tissue. At 24–72 hr post-inoculation i.v., the relative total parasite loads were significantly lower in the brain compared with lung and liver for two different mice strains (*Figure 2a, b and c*), as previously reported in different experimental approaches using oral and i.p. routes (*Kanatani et al., 2017*; *Zenner et al., 1998*; *Mordue et al., 2001*). Moreover, the restrictiveness of the different blood-CNS barriers is heterogeneous, especially in the meninges and CP (*Wilhelm et al., 2016*). To identify putative entry sites to the CNS, we classified the parasite foci, as defined under Methods, on the basis of (*i*) anatomical localization and (*ii*) in situ localization in relation to the adjacent vasculature (*Figure 2d and e*, *Figure 2—figure supplement 1*). Higher total numbers of parasite foci localized to parenchymal vessels compared with meningeal vessels and CP (*Figure 2f*). However, when the number of parasite foci was normalized to the relative tissue area, a significantly higher relative frequency was observed in the CP, but not in meningeal vessels (*Figure 2g*). The data corroborates that the vast majority of parasite foci localized to parenchymal tissue and its vasculature (*Figure 1*). However, the CP tissue contained a higher relative foci density compared with the parenchymal tissue and its vasculature. Altogether, this indicates that, while the vast majority of parasites localize to the cortical capillary bed, both the choroidal and meningeal blood-CNS barriers can serve as a point of entry for *T. gondii* to the CNS.

## *T. gondii* tachyzoites invade the brain parenchyma in the vicinity of capillaries

To analyze the progress of infection in situ around the parenchymal vasculature, the localization of foci with replicating *T. gondii* (GFP+) was related to the vascular marker EB and categorized as intravascular, perivascular or parenchymal localization (*Figure 3a*). Quantifiable numbers of replicating parasites were detected at 48 hr post-inoculation in the vasculature and an elevation in numbers was observed 72 hr post-inoculation with detection of parasites perivascularly and in the parenchyma (*Figure 3b*), in line with the elevation of total parasite loads in the brain (*Figure 2b and c*). A detailed analysis of the localization of *T. gondii* tachyzoites in relation to the vasculature (*Figure 3c and d*, *Videos 1–8*) revealed the distribution of parasite distances around parenchymal capillaries and indicated parenchymal localization (*Figure 3e*, *Figure 3—figure supplement 1a*). For randomly located parasite vacuoles, analyzes of distances to vasculature and of vascular diameter revealed a preferential vicinity to one capillary, related to the average distances to surrounding vasculature (*Figure 3f*, *Figure 3—figure supplement 2*). Localization of vacuoles containing replicating tachyzoites distantly from the blood tracer and vascular markers, and in the vicinity astrocyte and microglia markers, confirmed penetration to the parenchyma (*Figure 3g*, *Figure 3—figure supplement 1b*).

Because the mouse genetic background and the parasite lineage impact the cerebral pathology and the immune responses (*Schlüter and Barragan, 2019*), we assessed two additional mouse strains (CD1, BALB/c) and one additional parasite strain (ME49). A similar distribution of *T. gondii* was found around the parenchymal vasculature by days 3–4 post-inoculation in the different mouse strains (*Figure 3—figure supplement 1c-f*).

Next, to confirm the distribution of parasites in live brain tissue, freshly extracted whole brains from challenged mice were analyzed ex vivo by two-photon microscopy (*Videos 9 and 10*). Similar to fixed brain sections, the intravascular parasite localization predominated at 72 hr, with relatively lower numbers of tachyzoites in the perivascular space and parenchyma (*Figure 3h*). Furthermore, extravascular (perivascular and parenchymal) parasites localized to the vicinity of a blood vessel (*Figure 3i*) and both intravascular and extravascular parasites localized within or in the vicinity, respectively, of vessels

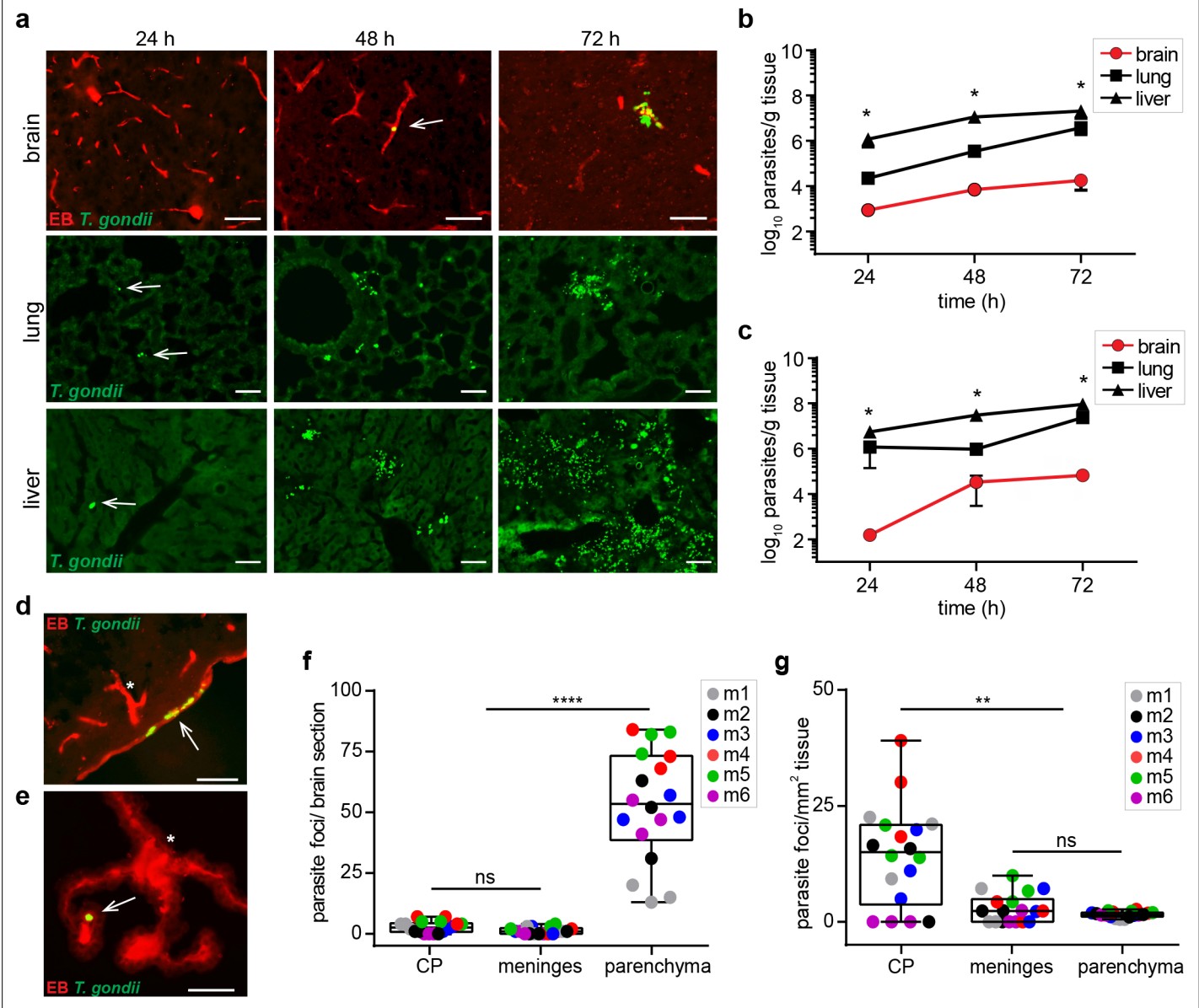

**Figure 2.** Parasite loads in peripheral organs and localization to vasculature in brain tissue. (**a, b, d-g**) C57BL/6 mice or (**c**) BALB/c mice were challenged i.v. with 3–10 × 10[6] *T. gondii* GFP-expressing tachyzoites (RH). (**a**) Representative epifluorescence micrographs of brain, lung and liver sections, respectively, at indicated time-points post-inoculation. Arrowheads indicate localization of GFP-expressing *T. gondii* tachyzoites at early time-points. Brain sections show intraluminal vascular tracer (EB) injected prior to organ extraction. Scale bars, 50 μm. (**b, c**) Parasite loads in brain, lung and liver tissue, respectively, at indicated time-points of C57BL/6 mice (**b**) or BALB/c mice (**c**) were quantified by plaquing assays. Data show mean ± SEM (n = 3 mice per time-point) *p < 0.05, Kruskal-Wallis test, followed by Dunn's post-hoc test. (**d, e**) Representative immunofluorescence micrographs of brain sections show intraluminal vascular staining (EB) and localization of GFP-expressing *T. gondii* tachyzoites in (**d**) meningeal vessels and (**e**) CP 72 hr post-inoculation. Arrows indicate parasite localization. Asterisks indicate sub-meningeal penetrating cortical vessel and characteristic morphology of CP (*Figure 2—figure supplement 1*), respectively. Scale bars, 50 μm. (**f, g**) Box-and-whisker dot plots show for (**f**) total numbers of parasite foci, defined as presence of replicating GFP[+] *T. gondii*, in CP, meninges and parenchyma, and (**g**) total parasite foci numbers related to tissue area (mm[2]), respectively, 72 hr post-inoculation. Medians with 25th to 75th percentile of datasets are shown. Whiskers mark 5th and 95th percentile. Each dot represents mean value per brain section in three consecutive brain sections per mouse from separate experiments (n = 6 mice). **p < 0.01; ****p < 0.0001; ns, non-significant, Kruskal-Wallis test, Dunn's post-hoc test.

The online version of this article includes the following figure supplement(s) for figure 2:

**Figure supplement 1.** Characterization of *T. gondii* infection in the choroid plexus.

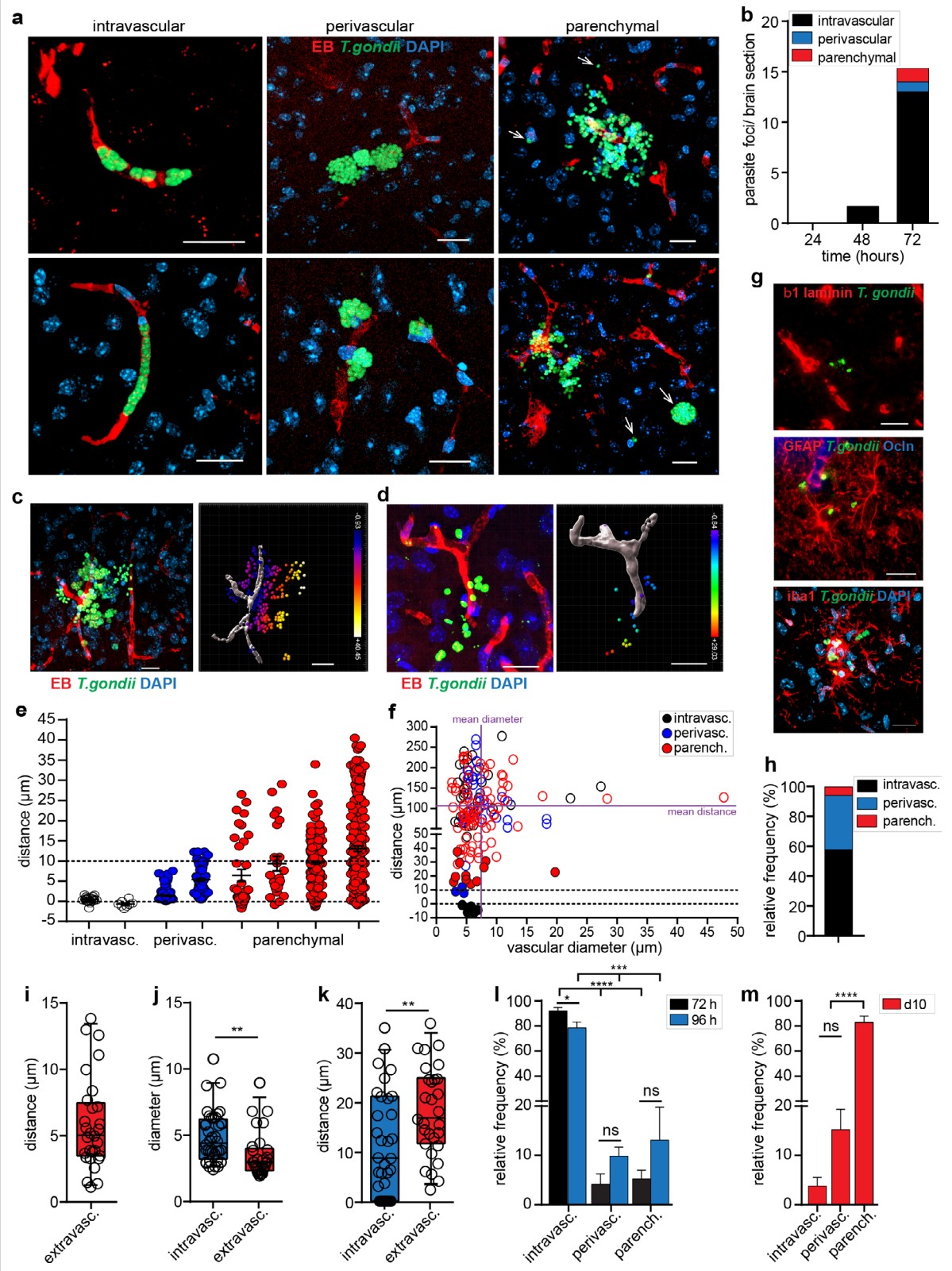

**Figure 3.** Distribution of parasite loads and invasion of the brain parenchyma by *T. gondii* tachyzoites. (**a–b**) C57BL/6 mice were challenged i.v. with 1–3 × 10⁶ *T. gondii* GFP-expressing tachyzoites (RH). (**a**) Representative confocal micrographs of parasite foci with replicating *T. gondii* (GFP⁺) categorized by association to the vasculature (EB): intravascular, perivascular and parenchymal localization, as indicated in Methods. DAPI indicates nuclear staining. Arrowheads indicate parasites located in the parenchyma. Scale bars, 20 µm. (**b**) Mean numbers of parasite foci (GFP⁺) detected per brain section at

*Figure 3 continued on next page*

*Figure 3 continued*

indicated time points post-inoculation. Data is from three sections per mouse from 3 mice per time point (n = 3 mice). (**c–k**) CD1 mice were challenged i.v. with 1–4 × 10⁶ *T. gondii* GFP-expressing tachyzoites (ME49) and sacrificed at 4 days post-infection. (**c, d**) Representative micrographs show the distribution of *T. gondii* tachyzoites (GFP⁺) around the vascular lumen (EB⁺) in challenged mice. For each micrograph, the corresponding 3D plot analysis shows individual GFP⁺ tachyzoite vacuoles as indicated in Methods. Color scales indicate Euclidian distances (µm) of tachyzoites to the nearest vascular lumen. Scale bars, 20 µm. (**e**) Distribution of Euclidian distances of *T. gondii* (GFP⁺) tachyzoites to the nearest capillary. Eight representative foci of challenged mice (day 4) are shown. In graph, each data point represents one tachyzoite (GFP⁺) and mean + SEM are indicated for each focus. Dashed lines indicate luminal center (0) and 10 µm distance to the nearest vascular lumen, respectively. (**f**) Distribution of Euclidian distances of randomly located *T. gondii* (GFP⁺) vacuoles to vasculature (EB⁺) in single fields of view (FOV). Filled circles indicate the distance to the nearest located blood vessel for intravascular, perivascular and parenchymal vacuoles, respectively. Unfilled circles indicate distances to vasculature in the same FOVs (*Figure 3—figure supplement 2*). Dashed lines indicate luminal center (0) and 10 µm distance to the nearest vascular lumen, respectively. Solid lines indicate mean diameter and mean distance, respectively, of vessels to randomly located vacuoles. Chi square test $X^2$(29, N = 166) = 64.96, p < 0.001. (**g**) Immunofluorescence staining of brain sections depict the parenchymal localization of *T. gondii* (GFP⁺) related to the BBB basement membrane marker β1 laminin, the astrocytic marker GFAP and the TJ marker occludin (Ocln) and, the microglia markers Iba1, respectively. DAPI indicates nuclear staining. Scale bars, 25 µm. (**h**) Relative distribution (%) of the localization of *T. gondii* tachyzoites assessed by two-photon microscopy in whole brains (day 3). n = 8 independent foci. (**i**) Distance of individual *T. gondii* tachyzoites to the nearest blood vessel (EB⁺). n = 10 independent foci. (**j, k**) Diameter of the nearest blood vessel (**j**) and distance to the nearest vascular branching (**k**) for intravascular and extravascular *T. gondii* tachyzoites, respectively. n = 69 tachyzoites. **p < 0.01; Student's t-test. (**i–k**) Medians with 25th to 75th percentile of datasets are shown. Whiskers mark 5th and 95th percentile. (**l, m**) Relative frequency (%) of parasite foci related to localization at indicated time points post-inoculation. CD1 Mice were inoculated with (**l**) high-dose or (**m**) low-dose *T. gondii* tachyzoites (ME49). For each condition, the mean + SEM from 3 to 10 sections per mouse from three independent experiments is shown (n = 3–6 mice). *p < 0.05; ***p < 0.001; ****p < 0.0001; ns, non-significant, one-way ANOVA, F (2, 10) = 79.19, followed by Tukey's multiple comparisons test.

The online version of this article includes the following figure supplement(s) for figure 3:

**Figure supplement 1.** Distribution of *T. gondii* in the intravascular and extravascular brain compartments in three mouse strains.

**Figure supplement 2.** Distances of *T. gondii* vacuoles to surrounding vasculature.

with <10 µm Ø (*Figure 3j*) and in the vicinity of vascular branching points (*Figure 3k*). Finally, the localization of *T. gondii* was monitored beyond 72 h. Importantly, while intravascular and perivascular parasite localization predominated days 1–4 post-inoculation (*Figure 3l*), parenchymal localization clearly predominated by day 10 (*Figure 3m*, *Figure 3—figure supplement 1g*). Of note, at all time-points, focal leakage of the EB blood tracer to the parenchyma was not observed, indicating absence of hemorrhage. Altogether, the data show that breaching of the BBB by *T. gondii* predominantly occurs at the level of parenchymal capillaries.

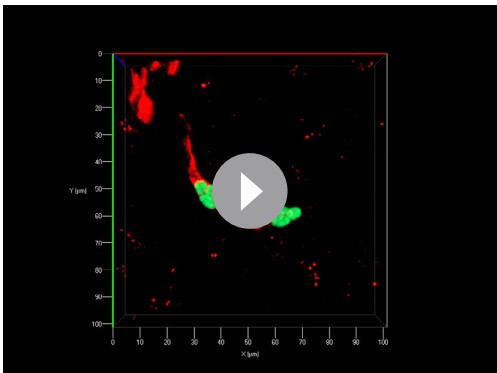

**Video 1.** Intravascular replication of *T. gondii* in brain microvasculature. Movie shows 3D projections of vacuoles with replicating *T. gondii* (GFP⁺, green) and association to the vasculature (EB⁺, red) as detailed in Methods.
https://elifesciences.org/articles/69182/figures#video1

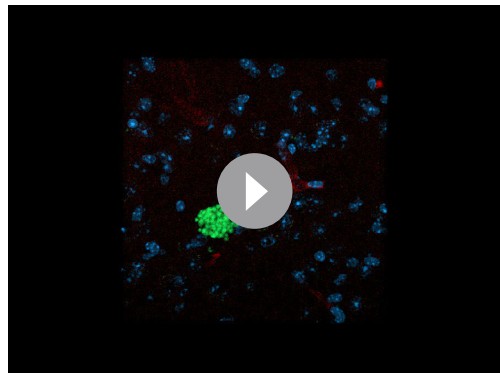

**Video 2.** Perivascular replication of *T. gondii*. Movie shows 3D projections of parasite foci with replicating *T. gondii* (GFP⁺) in close proximity to the vasculature (EB⁺, red) as detailed in Figure 3. Nuclei (blue) were stained with DAPI.
https://elifesciences.org/articles/69182/figures#video2

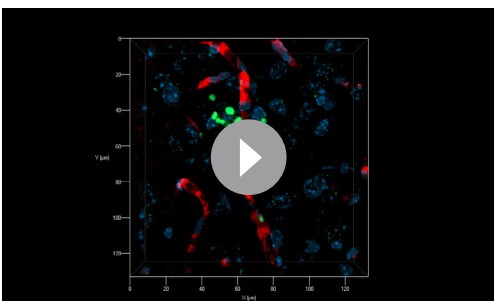

**Video 3.** Parenchymal localization of *T. gondii*. Movie shows 3D projections *T. gondii* tachyzoites (GFP⁺) and spatial relation to the vasculature (EB⁺, red).
https://elifesciences.org/articles/69182/figures#video3

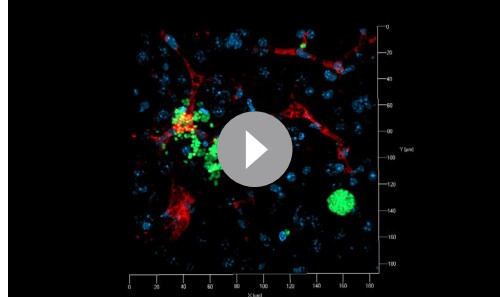

**Video 5.** Penetration to the parenchymal tissue by *T. gondii*. Movies show 3D projections of representative parasite foci with replicating *T. gondii* (GFP⁺) within the brain parenchyma (cell nuclei, DAPI/ blue) and association to the vasculature (EB⁺, red).
https://elifesciences.org/articles/69182/figures#video5

## Kinetics of leukocyte infiltration at parenchymal sites of *T. gondii* replication

Because parasite penetration to the parenchymal tissue may imply inflammatory responses, we assessed the presence of leukocytes at loci of *T. gondii* replication in the cerebral and pulmonary parenchyma. In the lung, a prominent leukocyte (CD45⁺) infiltration was rapidly observed in the vicinity of parasite foci by days 4–5 (*Figure 4a*). In contrast, the vast majority of parasite foci in the brain parenchyma stained negative for CD45 by days 4–5 post-inoculation, irrespective of parasite dose (*Figure 4a and b*). However, an abundant infiltration of CD45⁺ cells around parasite foci was evidenced by day 10 (*Figure 4b*, *Figure 4—figure supplement 1a*). Analyses by flow cytometry (*Figure 4—figure supplement 1b, c*) confirmed elevated CD45⁺ CD11b⁺/⁻ cell frequencies in lung by days 4–5 (*Figure 4c, d and f*) and modest elevations of CD45⁺ cells in the brain day 5 (*Figure 4c, d and e*). An elevation in CD45ʰⁱ CD11b⁺/⁻ cell frequencies in the brain by day 10 (*Figure 4d and e*) corroborated immunohistochemical findings. Additionally, a significant portion of *T. gondii* tachyzoites also co-localized with intravascular CD45⁺ cells (*Figure 4—figure supplement 2a, b*) and a portion of parasite-associated microglia (CD45ˡᵒʷ CD11b⁺) and CD45⁺ cells were retrieved from the brain tissue (*Figure 4—figure supplement 2c-j*). Altogether, the data indicate that initial parasite penetration to the parenchyma by day 4–5 occurs in absence of perivascular cuffing or a significant leukocyte infiltration, which is accentuated in the brain parenchyma by day10.

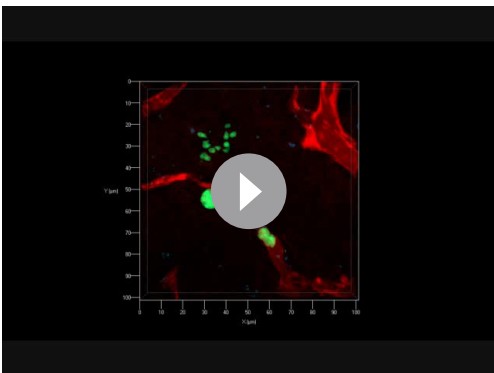

**Video 4.** Intravascular, perivascular, and parenchymal localization of *T. gondii*. Movie shows 3D projections of parasite foci with replicating *T. gondii* (GFP⁺) within a microvessel (EB⁺, red), in close proximity outside vasculature and distant location.
https://elifesciences.org/articles/69182/figures#video4

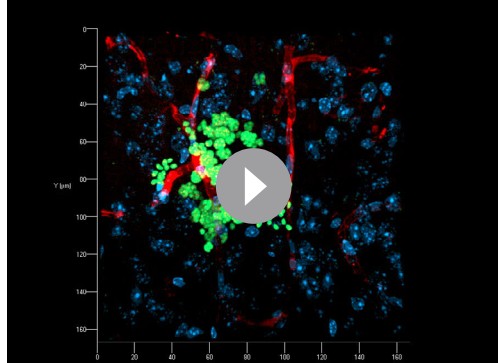

**Video 6.** Penetration to the parenchymal tissue by *T. gondii*. Movies show 3D projections of representative parasite foci with replicating *T. gondii* (GFP⁺) within the brain parenchyma (cell nuclei, DAPI/ blue) and association to the vasculature (EB⁺, red).
https://elifesciences.org/articles/69182/figures#video6

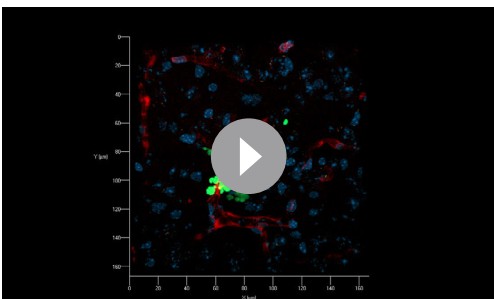

**Video 7.** Penetration to the parenchymal tissue by *T. gondii*. Movies show 3D projections of representative parasite foci with replicating *T. gondii* (GFP⁺) within the brain parenchyma (cell nuclei, DAPI/ blue) and association to the vasculature (EB⁺, red).
https://elifesciences.org/articles/69182/figures#video7

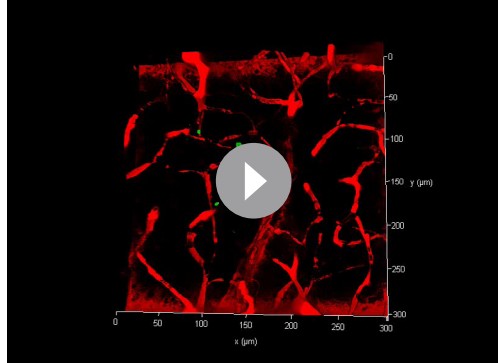

**Video 9.** Two-photon microscopy analyses of brain tissue ex vivo. Brains were extracted from *T. gondii*-challenged mice and immediately subjected to microcopy as indicated under Materials and methods. Movies show 3D projections of representative parasite foci with replicating *T. gondii* (GFP⁺) and spatial relation to the vasculature (EB⁺, red).
https://elifesciences.org/articles/69182/figures#video9

## Invasion of *T. gondii* to the brain parenchyma occurs with minimal perturbation of the vascular permeability

Next, we determined if challenge with *T. gondii* impacted the BBB restrictiveness to the permeability tracer EB, as detailed in Materials and methods (*Figure 5—figure supplement 1a, c*). High-dose inoculation of *T. gondii* significantly increased EB signal in the parenchyma, indicating elevated BBB permeability similar to the effects of LPS treatment (*Xaio et al., 2001*; *Figure 5a and b*). In contrast, a lower dose of *T. gondii* non-significantly affected permeability up to 5 days post-inoculation. Yet, at day 10, a significant elevation in permeability was observed (*Figure 5c and d*). Because early passage (24–72 hr) to the parenchyma (*Figure 3*) occurred seemingly in absence of measurable generalized elevations in permeability, we closely assessed the vascular permeability in the vicinity of parasite foci. Overall, non-significant differences in permeability were observed when comparing brain sections containing tachyzoites (GFP⁺) and sections with absence of tachyzoites (GFP⁻) at days 3 or 10 (*Figure 5e and f*). However, a subset of GFP⁺ vessels exhibited elevated EB signal, indicating focal or selective enhanced permeability in these vessels (*Figure 5f*). Areas with major EB intensity signal, indicative of hemorrhage (*Manaenko et al., 2011*), were absent throughout the study. We conclude that a generalized enhanced BBB permeability was only observed upon high-dose i.v. inoculation of *T. gondii*, while lower doses given i.p.

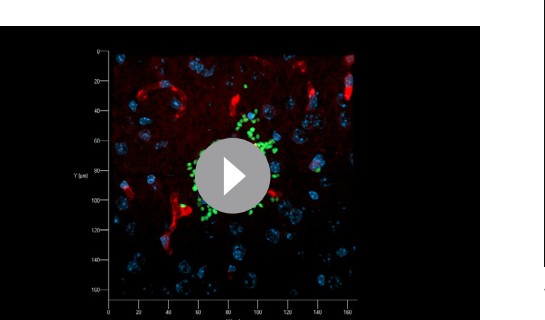

**Video 8.** Penetration to the parenchymal tissue by *T. gondii*. Movies show 3D projections of representative parasite foci with replicating *T. gondii* (GFP⁺) within the brain parenchyma (cell nuclei, DAPI/ blue) and association to the vasculature (EB⁺, red).
https://elifesciences.org/articles/69182/figures#video8

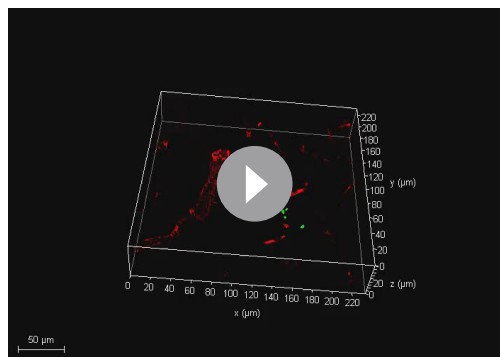

**Video 10.** Two-photon microscopy analyses of brain tissue ex vivo. Brains were extracted from *T. gondii*-challenged mice and immediately subjected to microcopy as indicated under Materials and methods. Movies show 3D projections of representative parasite foci with replicating *T. gondii* (GFP⁺) and spatial relation to the vasculature (EB⁺, red).
https://elifesciences.org/articles/69182/figures#video10

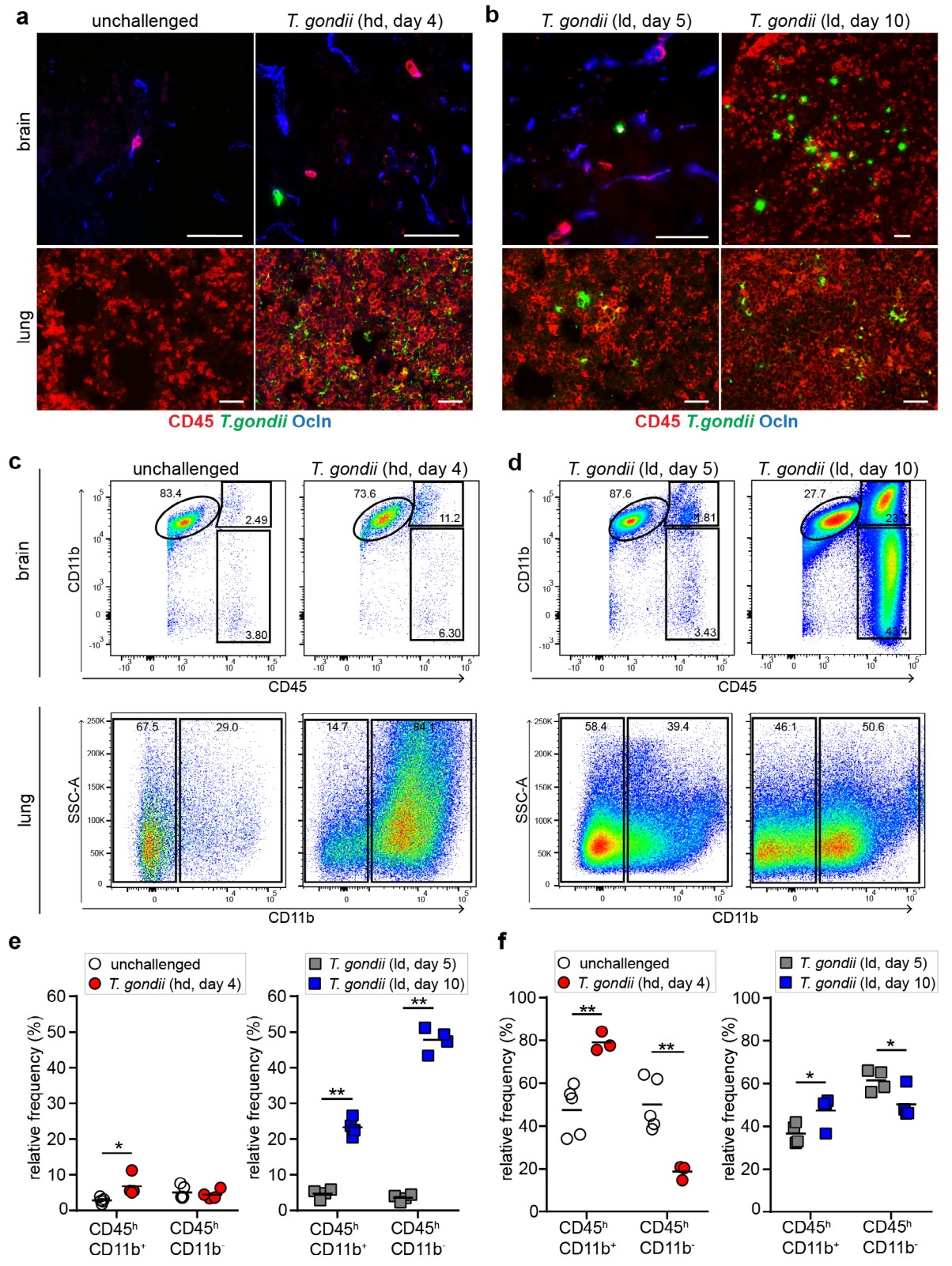

**Figure 4.** Kinetics of leukocyte infiltration in the brain parenchyma. CD1 mice were inoculated with high-dose (hd, 1 × 10⁶ i.v.) or low-dose (ld, 5 × 10⁴ i.p.) of *T. gondii* GFP-expressing tachyzoites (ME49) (**a, b**) Representative immunofluorescence micrographs of brain and lung cryosections from mice challenged with *T. gondii* (GFP⁺) at (**a**) high-dose or (**b**) low-dose, assessed after days 4, 5, or 10 post-inoculation. Sections were stained for CD45 and occludin (Ocln). Scale bars, 50 µm. (**c, d**) Flow cytometry analyses of brain and lung tissue homogenates labeled with anti-CD45 and anti-CD11b

*Figure 4 continued on next page*

*Figure 4 continued*

antibodies, with gating on CD45⁺ cells. Representative bivariate plots (brain) show subpopulations of leukocytes discriminated according to the expression of CD11b and CD45: microglia (CD11b⁺ CD45ˡᵒʷ), myeloid leukocytes (CD11b⁺ CD45ʰⁱ) and non-myeloid leukocytes (CD11b⁻ CD45ʰⁱ). Plots from lung tissue show CD11b⁻ and CD11b⁺ subpopulations of CD45⁺ cells. (**e, f**) Quantitative analyses of CD45⁺ immune cell subpopulations in the (e) brain and (f) lung. The relative mean frequency of the leukocyte subpopulations is indicated (n = 3–5 mice per condition). *p < 0.05; **p< 0.01; Student's t-test.

The online version of this article includes the following figure supplement(s) for figure 4:

**Figure supplement 1.** Leukocyte infiltration in the brain parenchyma and gating strategy for flow cytometry.

**Figure supplement 2.** Parasites associate with resident and infiltrating CD45⁺ cells in the brain.

generated overall non-significant differences in BBB permeability, with focal permeability elevations in the vicinity of parasite foci.

## Inflammatory responses in brain endothelium upon challenge with *T. gondii*

To address the putative endothelial responses to the presence of *T. gondii* in capillaries, we assessed the inflammatory responses of primary mouse brain endothelial cell (MBEC) monolayers, of purified brain microvessels and in vivo in mice. First, transcriptional analyses of MBECs challenged with *T. gondii* demonstrated upregulation of a number of inflammation-associated genes, similar to LPS treatment (*Figure 6a*). Second, isolated brain microvessels from *T. gondii*-infected mice (*Figure 6b*, *Figure 6—figure supplement 1a-c*), consistently presented upregulated transcriptional expression of the adhesion molecules and inflammation indicators ICAM-1, VCAM-1, E-selectin, and anti-inflammatory TIMP-1 (*Figure 6c and d*). Finally, an upregulation of ICAM-1 and VCAM-1 expression was confirmed in isolated brains from infected mice (*Figure 6e and f*).

## Effects of anti- and pro-inflammatory treatments on vascular permeability and penetration of *T. gondii* to the brain parenchyma

To address if inflammation was a determinant of *T. gondii* passage across the BBB, we quantified the invasion of *T. gondii* to the parenchyma in two situations (*i*) reduced inflammation by hydrocortisone treatment before and during infection and (*ii*) increased inflammation upon low-dose LPS treatment before and during infection. First, we confirmed that hydrocortisone treatment efficiently reduced pro-inflammatory IFN-γ and IL-12 responses upon *T. gondii* challenge (*Figure 7—figure supplement 1a-d*). Conversely, LPS-treated *T. gondii*-challenged mice maintained elevated IFN-γ and IL-12 responses with leukocytosis (*Figure 7—figure supplement 1a, c, e-k*). Next, we assessed the impact of the treatments on parasite loads. At day 6 post-inoculation, higher parasite loads were detected in brain, liver and spleen in hydrocortisone-treated challenged mice (*Figure 7a*) and in LPS-treated challenged mice (*Figure 7b*). Importantly, characterizations of parasite foci in brain tissue revealed dramatic differences between the treatments (*Figure 7c*). First, LPS treatment impacted the distribution of localization of parasite foci (*Figure 7d*) with elevated numbers of foci in the parenchyma (*Figure 7e*). Second, the parenchymal foci were significantly larger compared with foci in unchallenged or hydrocortisone-treated mice (*Figure 7f*). Finally, LPS-treated mice exhibited elevated BBB permeability (*Figure 7g*). Hydrocortisone treatment yielded minor or non-significant differences in parenchymal foci numbers and size, and BBB permeability (*Figure 7e–g*). At early time-points (day 3), infected mice untreated or treated with hydrocortisone or LPS exhibited non-significant differences in parasite loads and in the localization of parasites in parenchyma and the perivascular space (*Figure 7—figure supplement 2a-e*), indicating that the effects of hydrocortisone and LPS appeared posteriorly. We conclude that anti-inflammatory treatment of mice with hydrocortisone elevated total parasite loads with non-significant effects on vascular permeability, numbers and size of *T. gondii* foci in the brain parenchyma. In contrast, pro-inflammatory treatment with LPS elevated vascular BBB permeability, and increased numbers and size of parenchymal parasitic foci. Jointly, this indicated that both the inflammatory immune response and the restrictiveness of the BBB impacted the parasite loads in the brain parenchyma.

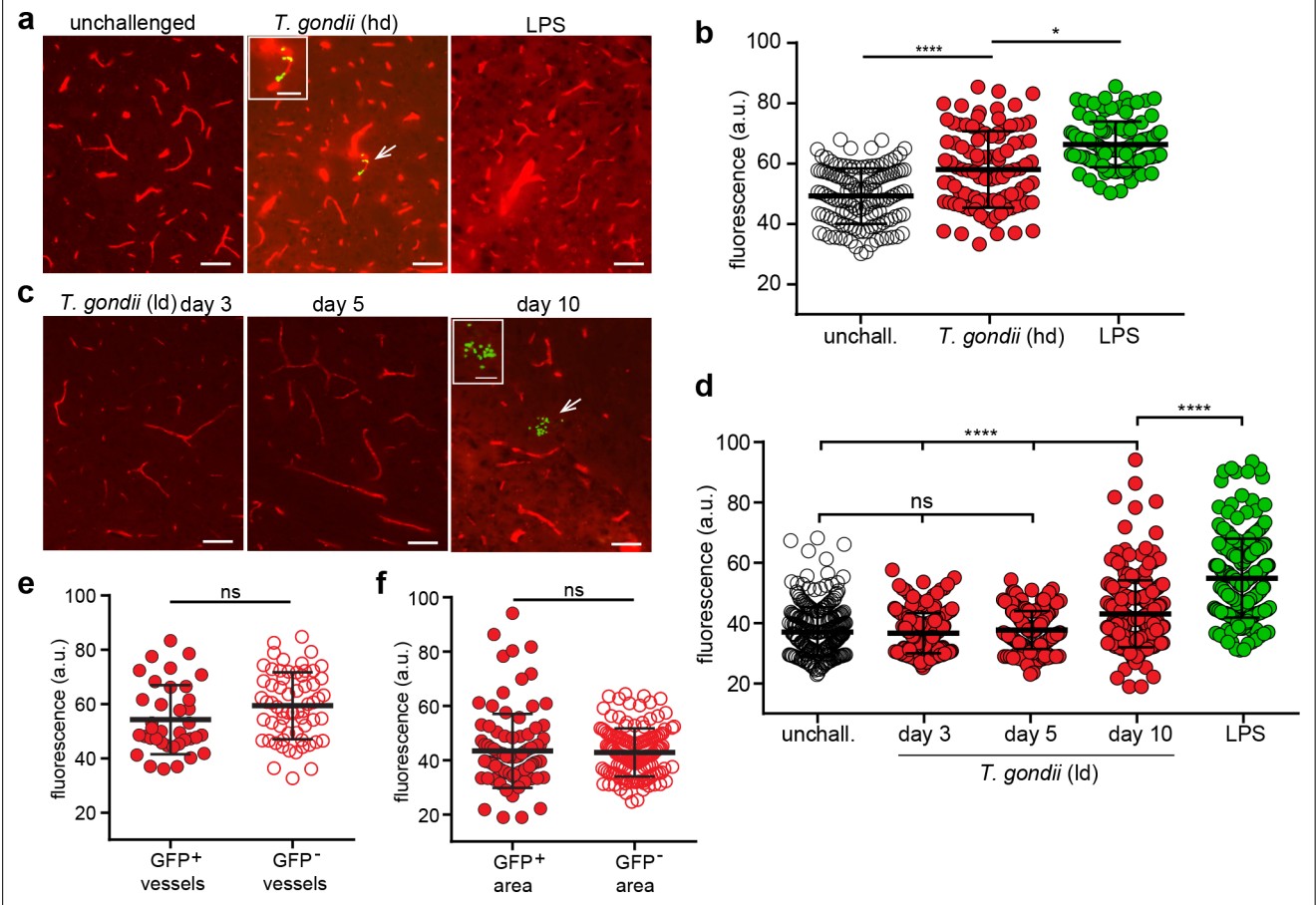

**Figure 5.** Impact of *T.gondii* challenge on the BBB permeability. (**a–b**) C57BL/6 mice were inoculated with high-dose (hd, 2–4 × 10^6 i.v. RH) or (**c–d**) with low-dose (ld, 5 × 10^4 i.p. ME49) of *T. gondii* GFP-expressing tachyzoites or treated with LPS (2 mg/kg, 6 hr). (**a**) Representative fluorescence micrographs of cortical sections show EB^+ signal from unchallenged, *T. gondii*-challenged (GFP^+) mice at day 3 post-inoculation and LPS-treated mice. Arrowhead indicates GFP signal, magnified in inset image. Scale bars, 50 μm, inset, 25 μm. (**b**) Quantification of the extravascular fluorescence intensity (arbitrary units, a.u.) with the EB tracer as described in Methods. Each dot in the graph represents the accumulated fluorescence intensity signal (EB^+) around one blood vessel. Data show mean ± SD from a total of 344 vessels (unchallenged 132, *T. gondii* 97, LPS 115) from three to four sections per mouse (n = 4–5 mice per condition). (**c**) Representative fluorescence micrographs of cortical sections show EB^+ signal as in (**a**) from *T. gondii*-challenged (ld, low-dose, ME49) mice at days 3, 5, and 10 post-inoculation. Arrowhead indicates GFP signal, magnified in inset image. Scale bars, 50 μm, inset image, 25 μm. (**d**) Quantification of the extravascular fluorescence intensity with the EB tracer as in (**b**). Data show mean ± SD from a total of 1205 vessels (unchallenged 280, *T. gondii* day3 214, day5 244, day10 234, LPS 233) from three sections per mouse from three independent experiments (n = 3–8 mice per condition). (**e, f**) Fluorescence intensity analyses with EB tracer signal in (**e**) GFP^+ and GFP^- cortical vessels from mice challenged with *T. gondii* (hd, day 3) or in (**f**) microvessels located within 100 μm distance to parasite foci (GFP^+) and fields of view without detectable foci (GFP^-) from mice challenged with *T. gondii* (ld, day 10). Data show mean ± SD from (**e**) 97 vessels (GFP^+ 37, GFP^- 60) and (**f**) 234 vessels (GFP^+ 96, GFP^- 138), respectively (n = 3–4 mice per condition). *p < 0.05; ****p < 0.0001; ns, non-significant. (**b, d**) Kruskal-Wallis test, Dunn's post-hoc test, (**e, f**) Mann-Whitney test.

The online version of this article includes the following figure supplement(s) for figure 5:

**Figure supplement 1.** Permeability analyses of cortical microvessels with vascular fluorescent tracer.

## Treatment with a FAK inhibitor leads to elevated parasite loads in the brain parenchyma

To determine the impact of the BBB integrity on parenchymal parasite loads, mice were treated with a second-generation inhibitor (defactinib, VS) (*Kanteti et al., 2018*; *Gerber et al., 2020*) of the intercellular junction regulator FAK. VS treatment led to significantly elevated parasite loads in the brain for both high-dose (hd) and low-dose (ld) *T. gondii* challenges (*Figure 8a, b and c*, *Figure 8—figure supplement 1a*), with an impact on parasite loads in peripheral organs. Detected IFN-γ/IL-12 responses in brain homogenate were in similar range as untreated, with variability between mice (*Figure 8—figure supplement 1b, c*). Importantly, the number of parasites in the intravascular and

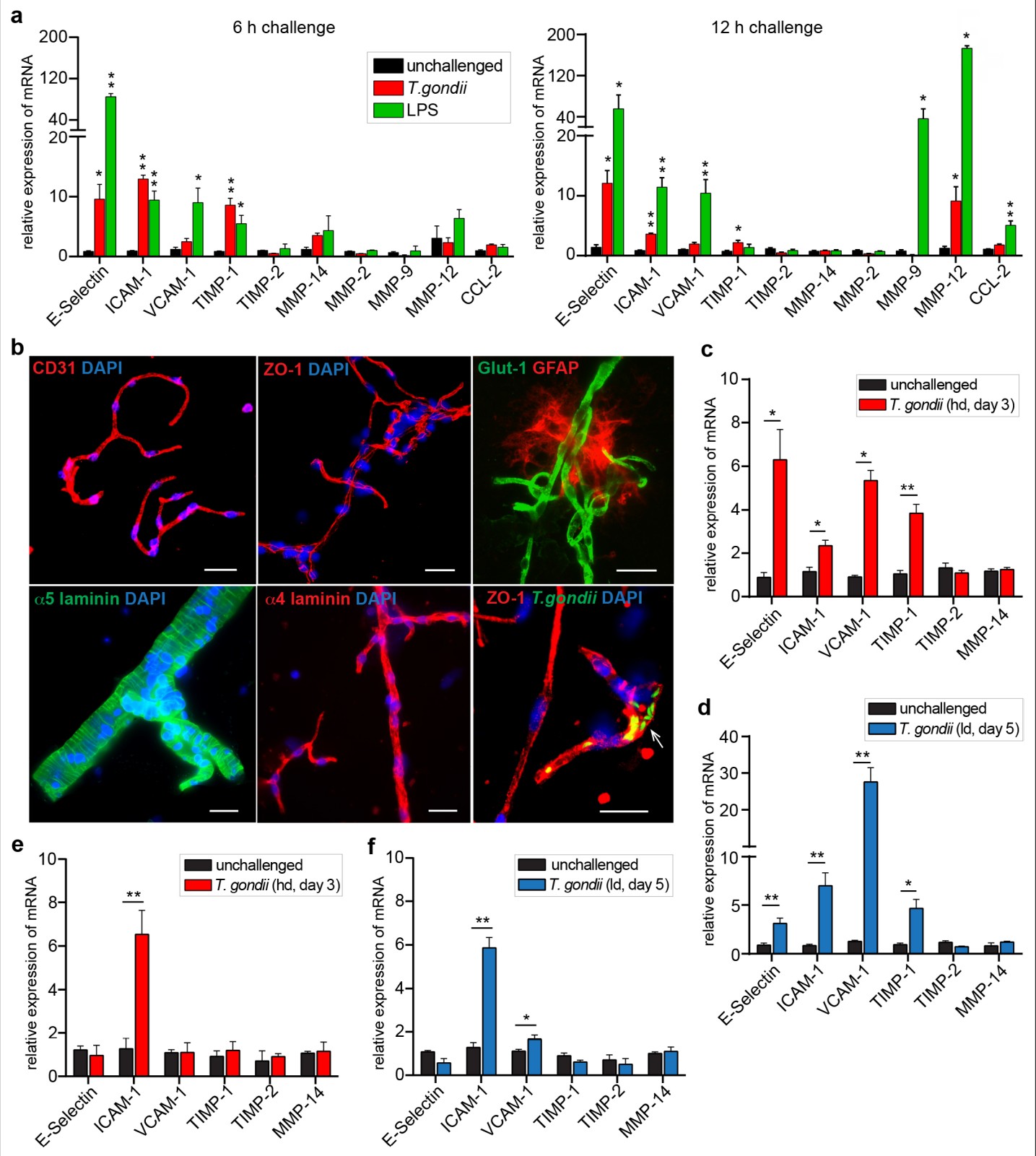

**Figure 6.** Inflammatory responses of brain endothelium upon *T. gondii* challenge. (**a**) Graphs show the relative mRNA expression, for indicated target genes, in primary MBECs challenged with *T. gondii* tachyzoites (RH, MOI4) or LPS (100 ng/ml) for 6 or 12 hr, respectively. Data show mean + SEM normalized to non-stimulated controls (unchallenged) from two independent MBEC purifications with 3–7 replicates/condition. (**b**) Representative immunofluorescence micrographs show brain microvessels isolated from unchallenged and *T. gondii* (RH)-challenged C57BL/6 mice stained with

*Figure 6 continued on next page*

*Figure 6 continued*

antibodies to the vascular BBB markers CD31/PECAM-1, ZO-1, Glut-1, α5/α4 laminins and the astrocytic marker GFAP. DAPI indicates nuclear staining. Arrowhead indicates *T. gondii* tachyzoites (GFP⁺) in association with the vasculature. Scale bars, 25 μm. (**c, d**) Relative mRNA expression, for indicated target genes, in purified brain microvessels of C57BL/6 mice challenged with *T. gondii*. (**c**) high-dose (hd, $4 \times 10^{6}$, RH) and (**d**) low-dose (ld, $5 \times 10^{4}$, ME49), assessed on days 3 and 5 post-inoculation, respectively. Data show mean + SEM gene expression from 3 to 4 mice per group normalized to non-infected controls (n = 3–4 mice). (**e, f**). Relative mRNA expression (mean + SEM), for indicated target genes, in brain homogenates from mice challenged as in (**c, d**) (n = 3–4).*p < 0.05; **p < 0.01; (**a**) one-way ANOVA followed by Dunnett's test, (**c–f**), Unpaired Student's t-test.

The online version of this article includes the following figure supplement(s) for figure 6:

**Figure supplement 1.** Immunostainings of isolated cortical microvessels.

---

perivascular space were significantly increased early during infection (*Figure 8d*). However, at day eight post-inoculation, VS-treated mice exhibited a significant increase in the number of parasite foci in the intravascular, perivascular and parenchymal space, related to untreated mice (*Figure 8e*). Moreover, the numbers and sizes of parasite foci were elevated (*Figure 8f and g*), indicating earlier penetration to the parenchyma upon VS-treatment. Leakage of the vascular tracer EB to the parenchyma was significantly increased in VS-treated mice compared to non-treated mice (*Figure 8h and i*), indicating increased permeability. Overall, we conclude that pharmacological inhibition of the vascular intercellular TJ regulator FAK increases BBB permeability and significantly elevates parasite loads in the brain parenchyma.

## Conditional deletion of endothelial FAK in mice impacts the passage of *T. gondii* to the brain parenchyma

The impact of pharmacological FAK antagonism on parenchymal parasite loads motivated a deletion of endothelial FAK-encoding *Ptk2* in mice. To generate endothelial cell-specific deletion of *Ptk2* (ECPTK2 cKO), *Ptk2^flox/flox^* mice were crossed with *Cdh5^cre+/-^* mice harboring a tamoxifen-inducible Cre recombinase under the regulation of vascular endothelial cadherin promoter. To confirm FAK deletion in genotyped mice (*Figure 9—figure supplement 1a*), fractions enriched in brain endothelial cells (MBECs) were isolated (*Ross et al., 2019*). Western blotting showed a significant reduction in total FAK protein expression in MBEC-enriched preparations from ECPTK2 cKO mice (*Figure 9a*), with undetectable changes of FAK expression in bone-marrow-derived cells (BMDCs) (*Figure 9—figure supplement 1b, c*). Tamoxifen-induced FAK deletion non-significantly impacted total parasite loads in brain tissue and in peripheral organs (*Figure 9b*), and BBB impermeability to EB was maintained (*Figure 9c*), consistent with the reported absence of detectable systemic derangement of vascular permeability (*Lee et al., 2010*; *Weis et al., 2008*). Importantly, a detailed analysis of parasite foci in the brain parenchyma (*Figure 9d*) detected elevated numbers (*Figure 9e*) and size (*Figure 9f*) of parasite foci in ECPTK2 cKO mice, related to ECPTK2 WT mice. Tamoxifen-treated cre control mice exhibited non-significant differences in peripheral parasite loads and parenchymal parasite foci numbers (*Figure 9—figure supplement 1d-f*). The data is consistent with a BBB dysregulative, but non-disruptive, effect of FAK deletion on MBECs (*Ross et al., 2019*). We conclude that endothelial FAK deletion impacts the passage of *T. gondii* to the brain parenchyma and yields elevated numbers of parasite foci.

Finally, we sought to determine the impact of FAK on the passage of a non-replicating uracil-auxotroph *T. gondii* mutant (CPS) (*Fox and Bzik, 2002*), shown to infect brain endothelium in mice (*Konradt et al., 2016*). First, we determined the invasive phenotype of the CPS mutant across polarized cellular monolayers with moderate-to-high (bEnd.3) and high (Caco2) transcellular electrical resistance (TCER), in presence and absence of uracil. Interestingly, freshly egressed CPS tachyzoites transmigrated across the polarized monolayers, with a non-significant reduction of transmigration frequencies in absence of uracil throughout the assay (16–18 h) and with relative lower transmigration frequencies across monolayers with high TCER (*Figure 9g*). The TCER values and the permeability to a low-molecular-weight tracer were monitored throughout the assay (*Figure 9—figure supplement 2a, b, c*), in accordance with (*Ross et al., 2019*). In sharp contrast, upon uracil deprivation ≥24 hr, freshly egressed CPS tachyzoites dramatically reduced their transmigration frequencies (*Figure 9h*). Similarly, gliding motility analyses confirmed a significant reduction of gliding trail lengths by freshly egressed CPS tachyzoites, which was accentuated in absence of uracil (*Figure 9i*). In vitro, CPS tachyzoites invaded the polarized cell monolayers over time in absence of replication (*Figure 9—figure supplement 2d*),

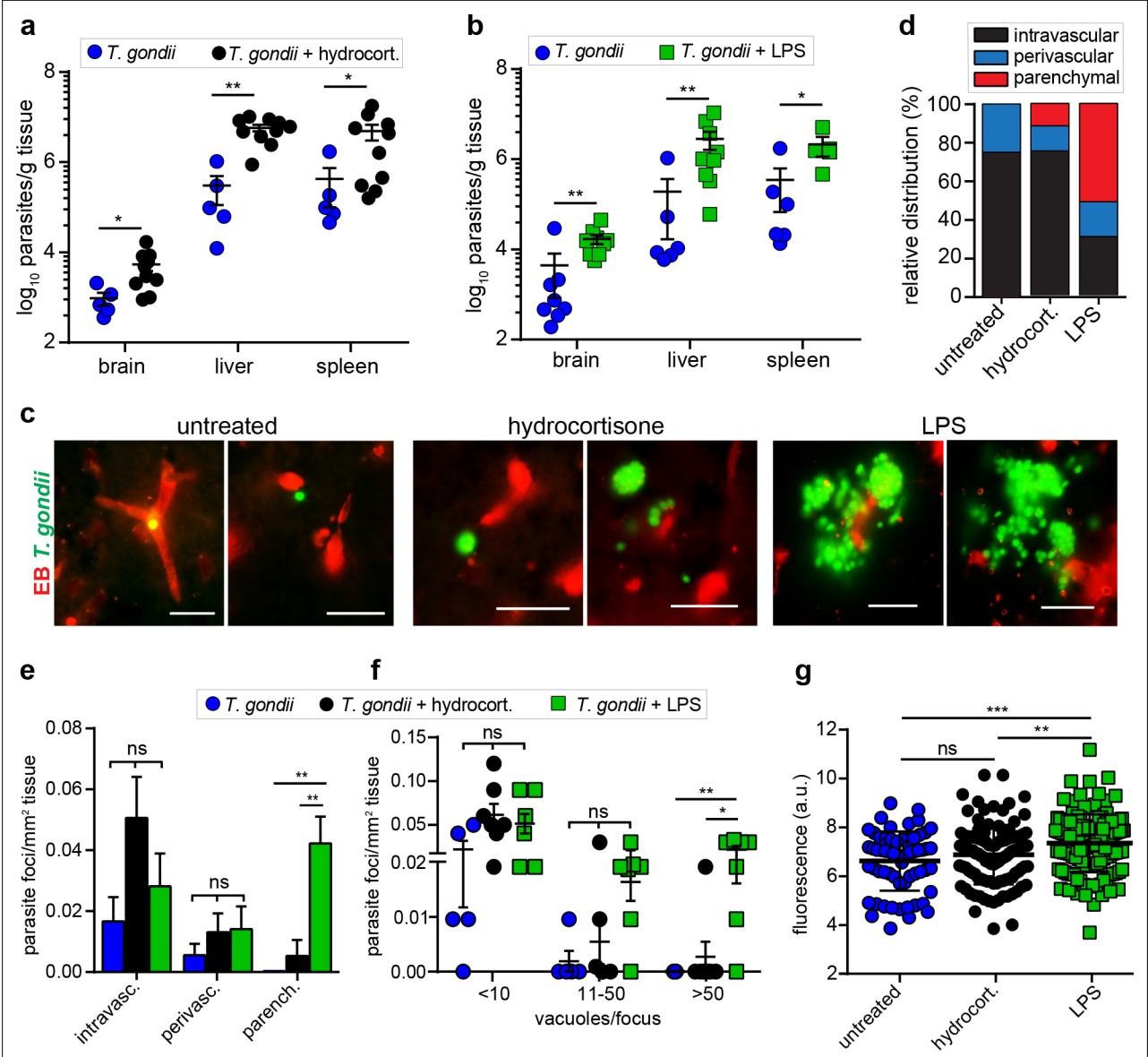

**Figure 7.** Modulation of parasite loads in the brain by anti-inflammatory and pro-inflammatory treatments. CD1 mice were inoculated with *T. gondii* GFP-expressing tachyzoites (ME49, 5 × 10⁴ i.p.). (**a, b**) Quantification of parasite loads by plaquing assays in brain, liver, and spleen of mice challenged with *T. gondii* (ld, day 6) and treated with (**a**) hydrocortisone or (**b**) LPS, as indicated under Materials and methods. Data show mean ± SEM (n = 5–10 mice per condition) *p < 0.05, **p < 0.01, Mann-Whitney test. (**c**) Representative micrographs of cortical sections from mice treated as in (**a, b**) show *T. gondii* (GFP+) and vascular tracer (EB). Scale bars, 25 μm. (**d**) Bar graph shows the relative proportion of intravascular, perivascular and parenchymal parasite foci in mice treated as in (**a, b**). (**e**) Relative numbers of parasite foci per mm² tissue related to intravascular, perivascular, and parenchymal localization, respectively, in mice treated as in (**a, b**). Data show mean + SEM **p < 0.01; ns, non-significant, Kruskal-Wallis test, Dunn's post-hoc test. (**f**) Relative numbers of parasite foci per mm² tissue related to foci size. Foci sizes were determined by counting number of vacuoles (<10, 11–50 or >50 vacuoles/focus) containing replicating tachyzoites (two or more / vacuole). Data show means ± SEM *p < 0.05; **p < 0.01; ns, non-significant, Kruskal-Wallis test, Dunn's post-hoc test. (**g**) Quantification of the extravascular fluorescence intensity (arbitrary units, a.u.) with the EB tracer in mice treated as in (**a, b**). Each dot represents the accumulated fluorescence intensity signal (EB⁺) around one blood vessel. Data show mean ± SD from a total of 321 vessels (untreated 64, hydrocortisone 111, LPS 146). (**d-g**) Data are from two independent experiments; untreated condition (n = 5 mice), hydrocortisone (n = 7 mice) and LPS (n = 7 mice).

The online version of this article includes the following figure supplement(s) for figure 7:

**Figure supplement 1.** Cytokine (IFN-γ, IL-12p40) and leukocyte responses to anti-inflammatory and pro-inflammatory treatments.

**Figure supplement 2.** Parasite loads in the brain upon anti-inflammatory and pro-inflammatory treatments at early time-points.

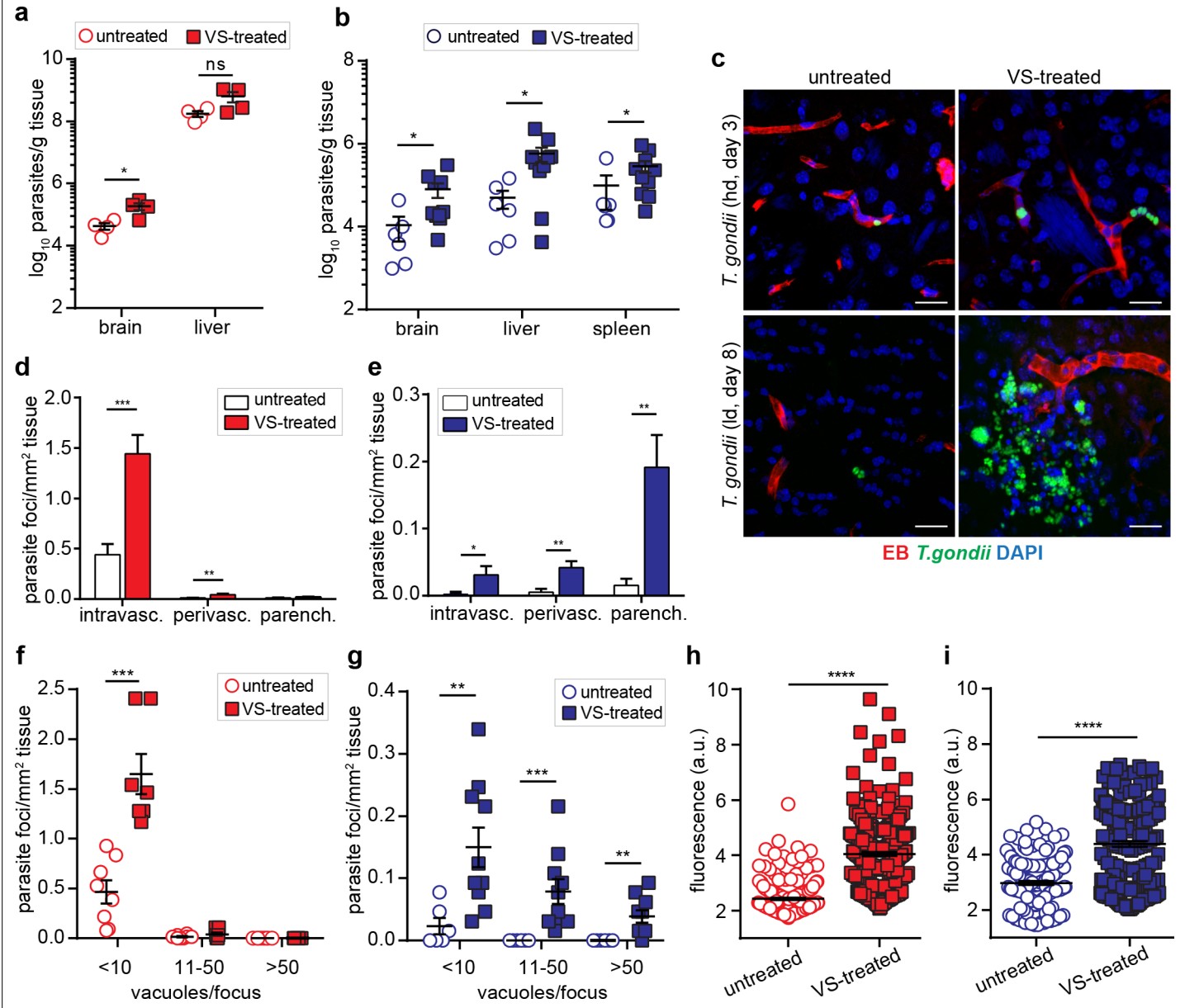

**Figure 8.** Modulation of parasite loads and BBB permeability by inhibition of FAK. CD1 mice were inoculated with high-dose (hd, $3 \times 10^6$ i.v., RH) or low-dose (ld, $5 \times 10^4$ i.p. ME49) of *T. gondii* GFP-expressing tachyzoites. (**a, b**) Quantification of parasite loads by plaquing assays in brain, liver and spleen of mice treated with VS-inhibitor and challenged with *T. gondii* at (**a**) high-dose or (**b**) low dose, assessed at days 3 or 8 post-inoculation, respectively. Data show mean + SEM from three independent experiments (n = 4–10 mice per group). (c)Representative micrographs of cortical sections from mice treated as in (**a, b**) show *T. gondii* (GFP+) and vascular tracer (EB). Scale bars, 25 µm. (**d, e**) Bar graph shows the relative proportion of intravascular, perivascular and parenchymal parasite foci in mice treated with VS-inhibitor and challenged with *T. gondii* at (d) high-dose or (e) low-dose, assessed at day three or eight post-inoculation, respectively. Data are from four independent experiments (n = 6–10 mice per group). (**f, g**) Relative numbers of parasite foci per mm² tissue related to foci size, in mice treated as in (a, b), respectively. Foci sizes were determined by counting number of vacuoles ( < 10, 11–50 or >50 vacuoles/focus) containing replicating tachyzoites. Data show mean + SEM from four independent experiments (n = 6–10 mice per group). (**h, i**) Quantification of the extravascular fluorescence intensity (arbitrary units, a.u.) with the EB tracer in mice treated as in (a, b). Each dot represents the accumulated fluorescence intensity signal (EB+) around one blood vessel. Data show mean ± SD from a total of (h) 517 vessels (untreated 217, VS-treated 300) and (i) 374 vessels (untreated 153, VS-treated 221) from two independent experiments (n = 6–7 mice per condition). *p< 0.05; **p < 0.01; ***p < 0.001; ****p < 0.0001; ns, non-significant, (**a,d,f**) Unpaired Student's t-test, (**b,e,g,h,i**), Mann-Whitney test.

The online version of this article includes the following figure supplement(s) for figure 8:

**Figure supplement 1.** Cytokine responses of mice treated with FAK inhibitor.

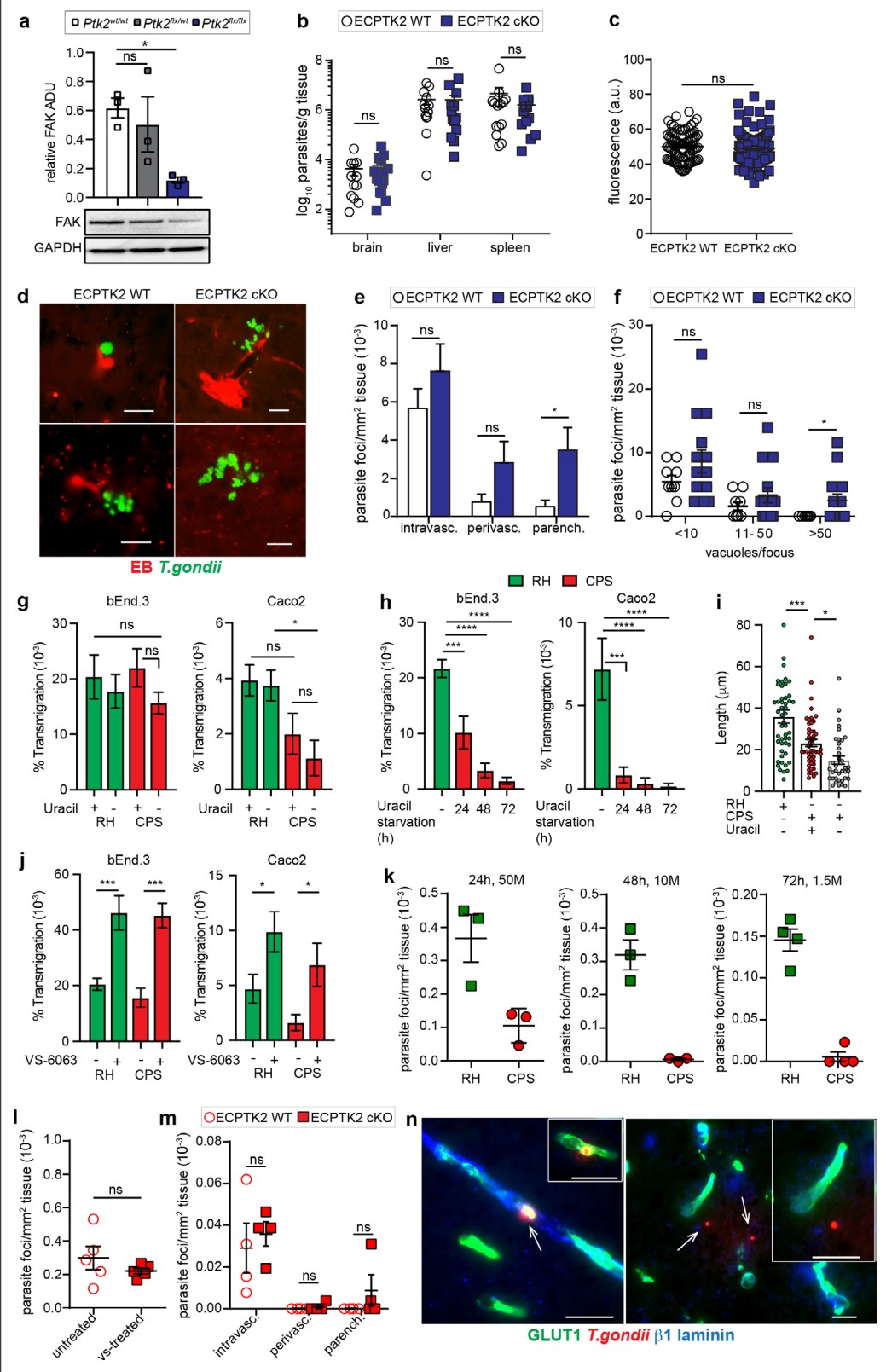

**Figure 9.** Characterizations of parasite foci in the brain parenchyma of ECPTK2 cKO mice challenged with wildtype (RH) and uracil-auxotroph *T. gondii* (CPS). (**a**) Western blot analysis of total FAK protein expression in endothelial cells derived from *Cdh5*[cre+/-] C57BL/6 mice that were *Ptk2*[wt/wt], *Ptk2*[flox/wt], or *Ptk2*[flox/flox]. Graph shows FAK expression normalized to GAPDH expression. ADU, arbitrary densitometry unit. n = 3 biological replicates from independent

*Figure 9 continued on next page*

*Figure 9 continued*

blots. (**b**) Quantification of parasite loads by plaquing assays in brain, liver and spleen of ECPTK2 WT (*Ptk2^flox/flox^Cdh5^cre-/-^*) and ECPTK2 cKO (*Ptk2^flox/flox^Cdh5^cre+/-^*) C57BL/6 mice, respectively, challenged with *T. gondii* (ME49, 2,5–5 × 10⁴), assessed at day 6 post-inoculation. Data show mean + SEM from three independent experiments (n = 14–15 mice per group). (**c**) Quantification of the extravascular fluorescence intensity (arbitrary units, a.u.) with the EB tracer in mice treated as in (**b**). Data show mean ± SEM from a total of 86–100 vessels from two independent experiments (n = 4 mice per condition). (**d**) Representative micrographs of cortical sections from mice treated as in (**b**) show *T. gondii* (GFP⁺) and vascular tracer (EB). Scale bars, 25 µm. (**e**) Bar graph shows the relative proportion of intravascular, perivascular and parenchymal parasite foci in ECPTK2 WT and ECPTK2 cKO mice challenged with *T. gondii* as in (**b**) assessed at day 6 post-inoculation, respectively. Data are from three independent experiments (n = 11–14 mice per group). (**f**) Relative numbers of parasite foci per mm² tissue related to foci size, in mice treated as in (**b**), respectively. Foci sizes were determined by counting number of vacuoles ( < 10, 11–50 or >50 vacuoles/focus) containing replicating tachyzoites. Data show mean ± SEM from three independent experiments (n = 11–14 mice per group). (**g**) Transmigration frequencies of freshly egressed *T. gondii* tachyzoites (RH, CPS) across bEnd.3 or Caco2 polarized monolayers, in presence or absence of uracil, are shown as percentage (%) of tachyzoites added in the upper well and assessed by plaquing assays as described under Methods. Data show mean ± SEM from four independent experiments performed in duplicate. (**h**) Transmigration frequencies of freshly egressed CPS tachyzoites grown in absence of uracil for 24, 48, and 72 hr, respectively. Data show mean ± SEM from three independent experiments performed in duplicate (**i**) Gliding motility analyses of freshly egressed RH and CPS tachyzoites, in presence or absence of uracil. Compiled data show individual trail lengths (n = 45–50) and mean + SEM from three independent experiments. (**j**) Transmigration frequencies of *T. gondii* tachyzoites (RH, CPS) as in (**g**) in presence or absence of FAK inhibitor (VS-6063). Data show mean ± SEM from 3 (bEnd.3) and 4 (Caco2) independent experiments performed in duplicate. (**k**) Relative parasite frequency in the brain after co-infection in CD1 mice. Each mouse was co-inoculated i.v. with 5 × 10⁷ (50 M), 1 × 10⁷ (10 M) or 1.5 × 10⁶ (1.5 M) freshly egressed RH and an equal number of freshly egressed CPS tachyzoites, respectively. Consecutive coronal brain sections were assessed for *T. gondii* after 24, 48, or 72 hr, respectively, as detailed in (*Figure 9—figure supplement 3a-d*). Data show mean ± SEM from 3 to 4 mice per experiment and condition. (**l, m**) Parasite loads in the brains of CD1 mice treated with FAK inhibitor (**l**) or ECPTK2 cKO mice (**m**), respectively, challenged with freshly egressed CPS tachyzoites (2 × 10⁸, i.v.). Parasite loads were assessed as in (**k**). Data show mean + SEM from 2 experiments and 5 mice per condition. (**n**) Representative micrographs from brain sections of C57BL/6 mice challenged with CPS (2 × 10⁸, i.v.) and stained for endothelial marker GLUT1, the basal lamina markerβ1 laminin and anti-*T. gondii* antibodies followed by secondary antibody Alexa 488/405/594, respectively. Left: micrograph and inset depict intravascular CPS tachyzoites. Right: micrograph illustrates extravascular localization of CPS. Inset represents magnification of area of interest. Arrows indicate *T. gondii* tachyzoites. Scale bars, 25 µm (*Figure 9—figure supplement 3e*). *p< 0.05; **p < 0.01; ***p < 0.001; ****p < 0.0001; ns, non-significant (**a, g–j**) one-way ANOVA followed by Sidaks post-hoc test, (**b, c, e, f, l, m**), Mann-Whitney test.

The online version of this article includes the following source data and figure supplement(s) for figure 9:

**Source data 1.** Western blots corresponding to *Figure 9a*.

**Figure supplement 1.** Genotyping, FAK expression in tissue of ECPTK2 cKO mice and characterizations of parasite loads and foci of Cre control mice.

**Figure supplement 1—source data 1.** Western blots corresponding to *Figure 9—figure supplement 1b,c*.

**Figure supplement 2.** Assessments of polarization and barrier integrity for transmigration assays, and replication in monolayers.

**Figure supplement 3.** Parasite loads in brain a peripheral organs upon challenge with the non-replicating CPS line.

with intracellular parasite localization as described (*Fox and Bzik, 2002*). Importantly, in presence of FAK inhibitor, the relative transmigration frequencies of freshly egressed CPS tachyzoites increased significantly, proportionally similar to RH (~2–3 fold) (*Figure 9j*). In vivo, co-challenge of mice with CPS and RH at highest doses tolerated by mice 24, 48, or 72 hr, respectively, showed dramatically low parasite loads for CPS in brain and other organs, compared with RH (*Figure 9k*, *Figure 9—figure supplement 3a*). Partly contrasting with in vitro data, non-significant differences in parasite loads were detected upon pharmacological FAK inhibition or upon challenge of ECPTK2 cKO mice with CPS (*Figure 9l and m*). For CPS-challenged mice, staining with endothelial cell and basal lamina markers confirmed scarce invasion events of single CPS tachyzoites, encompassing both intravascular localization and extravascular localization in the parenchyma (*Figure 9n*, *Figure 9—figure supplement 3e*, *Video 11*). The low total numbers of CPS parasites retrieved in brain precluded direct

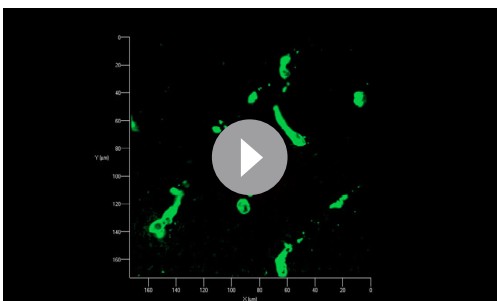

**Video 11.** Localization of tachyzoites (CPS) in brain tissue. Movie shows 3D projections of brain section from C57BL/6 mice challenged with freshly egressed CPS tachyzoites (2 × 10⁸, i.v.) and stained for endothelial marker GLUT1 (green) and anti-*T. gondii* followed by secondary antibody Alexa 594 (red).
https://elifesciences.org/articles/69182/figures#video11

comparisons with RH for translocation events or relative quantifications of passage across the BBB versus invasion of endothelial cells for this mutant line. We conclude that non-replicating CPS tachyzoites transmigrate across polarized mono-layers in a TCER-related fashion in vitro and that FAK inhibition elevates transmigration frequency. Upon uracil deprivation, the CPS line rapidly exhibits reduced gliding motility and transmigration frequency across polarized monolayers, compared with the parental strain RH. In vivo, low numbers of CPS parasites are retrieved in tissues hours after inoculation, indicating generally reduced invasion events and a rapid elimination of this parasite mutant in vivo.

## Discussion

Passage across the cellular blood-CNS barriers is a prerequisite for the establishment of latent cerebral toxoplasmosis in humans and rodents. Here, we studied the crucial interaction of *T. gondii* with the brain vasculature in mice.

Our study establishes that the principal point of entry of *T. gondii* from the blood circulation into the brain parenchyma is at the level of cortical capillaries, similarly in three separate mouse strains and for prototypic *T. gondii* type I and type II strains. The data evidence preferential localization of parasites to capillaries with low α5 laminin/P-glycoprotein expression ratio, a relative less frequent localization to post-capillary venules, and scarce or absence of localization to penetrating arterioles, pre-capillary arterioles and penetrating venules.

We report that the BBB integrity is paramount to restrict *T. gondii* infection of the brain paren-chyma. First, from day one post-inoculation, the accumulation of *T. gondii* tachyzoites in the brain was dramatically inferior to that observed in lung and liver, indicating a relative lower permissiveness of the brain vasculature to adhesion or invasive events by *T. gondii*. Second, the initial vascular localization with replicating parasites within brain endothelium (*Konradt et al., 2016*), was rapidly accompanied by appearance of vacuoles with replicating parasites in the perivascular spaces and in the cerebral parenchyma. Third, perturbations of BBB permeability by pro-inflammatory, anti-inflammatory stimuli or high doses of *T. gondii* resulted in elevated parasite loads in the CNS. Finally, pharmacological inhi-bition or induced gene ablation of the endothelial BBB intercellular junction regulator FAK facilitated parasite passage to the brain parenchyma.

In natural oral infections, relatively low numbers of *T. gondii* cysts ultimately form in the CNS indi-cating that passage across the BBB is a relatively rare event, compared with the initial parasite loads in the parenchyma of other peripheral organs, for example, lungs, liver, spleen, or in musculature. We validated an experimental approach that allowed controlled assessment of *T. gondii* localization in the CNS after intraperitoneal or intravenous inoculation and that yielded parasite load distributions comparable with oral and other intraperitoneal infection models (*Konradt et al., 2016*; *Kanatani et al., 2017*; *Zenner et al., 1998*; *Mordue et al., 2001*). While the approaches utilized in this and other studies permit reproducible quantifications, they may also entail limitations. An impact on the relative localization of parasites or kinetics of passage cannot be excluded. However, similar BBB localization and invasion patterns were observed for both inoculation routes, irrespective of mouse strain, parasite strain or dose. Importantly, for the vast majority of parasite foci, the clear vicinity to one capillary indicated the putative translocation site to the brain parenchyma. Further, the observa-tion of perivascular parasite vacuoles beneath the vascular endothelium is interesting and the identi-fication of the hosting cells remains to be investigated.

Our data show that *T. gondii* tachyzoites very early (1–3 dpi) interact primarily with the endothelium of cortical capillaries, confirming and extending the earlier reported replication of parasites in brain endothelium at later timepoints (7–9 dpi) (*Konradt et al., 2016*). This localization contrasts with find-ings in cerebral malaria, where parasitized erythrocytes preferentially bind to post-capillary venules and larger venules with components of inflammatory leukocytes (*Haldar et al., 2007*; *Nacer et al.,*

*2014*). The capillary bed offers the largest hemodynamic resistance to the cortical blood supply and has lower flow speeds (*Gould et al., 2017*). Hypothetically, this together with the restricted diameter of capillaries may facilitate the adhesion, invasion and crossing of *T. gondii* tachyzoites. In vitro, ICAM-1 and integrins are implicated in adhesion and transmigration of *T. gondii* tachyzoites (*Barragan et al., 2005*) and of parasitized leukocytes (*Ueno et al., 2014*; *Ross et al., 2021*). Among other inflammatory parameters and cell adhesion molecules (CAMs), here we report elevated expression of ICAM-1 and VCAM-1 in infected brains and, specifically, in purified brain microvessels from challenged mice. Jointly with the finding of elevated TIMP-1 expression in microvessels, and its recently attributed functions in maintenance of FAK-related BBB integrity (*Tang et al., 2020*) and *T. gondii* dissemination (*Ólafsson et al., 2018*; *Ólafsson et al., 2019*), future investigations need to address the implication of CAMs in endothelial and parenchymal invasion.

We provide evidence that *T. gondii* can invade the CNS across the choroid plexus (CP) and meningeal vessels, albeit in lower relative numbers compared with total parasite numbers entering across the extensive parenchymal vascular network. However, taking into account the relative smaller endothelial cell surface area and tissue volume of the CP and meningeal vessels related to cortical vasculature, the data indicate that passage is facilitated at the CP. Hypothetically, this may implicate the fenestrated choroidal endothelium, a known *locus minoris resistentiae* for brain invasion by the causative agent of sleeping sickness *Trypanosoma brucei* (*Schultzberg et al., 1988*). Similarly, bacteria and viruses can translocate across meninges and CP (*Schwerk et al., 2015*) or by disrupting the parenchymal BBB (*Doran et al., 2016*; *Spindler and Hsu, 2012*). The data is also in line with the reported localization of *T. gondii* reactivation foci in grey matter, with few reactivation foci in the CP and the meninges (*Dellacasa-Lindberg et al., 2007*). Further, human cerebral toxoplasmosis is associated with encephalitis and seldomly associated with meningeal inflammation (*Schlüter and Barragan, 2019*), while detection of *T. gondii* in the cerebrospinal fluid is used in clinical practice (*Mesquita et al., 2010*). Jointly, this indicates that, while parenchymal invasion occurs primarily across cortical capillaries, passage across the CP and meningeal vessels provides *T. gondii* access to the cerebrospinal fluid circulatory system, which may contribute to parasite dissemination within the CNS.

We report that early passage of *T. gondii* to the brain parenchyma occurs in absence of measurable BBB leakage, perivascular cuffing or prominent leukocyte infiltration, similarly in different mouse strains and for *T. gondii* type I and type II strains. In fact, measurable leukocyte infiltration in the vicinity of parasite foci was absent early during infection (< day 5). In contrast, infiltration was abundant by day 10, in line with recent findings (*Schneider et al., 2019*). Thus, perivascular cuffing or significant leukocyte infiltration does not appear to be a requisite for initial *T. gondii* passage to the parenchyma (< day 5). However, an inflammatory activation of the BBB endothelium by the infection was evidenced by elevated expression of CAMs but also inflammation-dampening TIMP-1 (*Ólafsson et al., 2018*; *Ólafsson et al., 2019*). Pro-inflammatory pre-treatment with LPS yielded elevations of parenchymal parasite loads and significantly larger cerebral parasite foci. This indicates that initial parasite passage was facilitated by LPS-induced increased BBB permeability (*Banks et al., 2015*), which may implicate regulation via FAK (*Guo et al., 2015*), and by exacerbated inflammation (*Haruwaka et al., 2019*). Conversely, anti-inflammatory pre-treatment with hydrocortisone led to elevated total parasite loads in the brain but also in other organs, likely secondary to a hampered immune response. Jointly, though these treatments have pleiotropic effects with a likely impact on parasitemia, the data is indicative that passage of *T. gondii* across the BBB is not only restricted by the inflammatory response. Along these lines, unnaturally high doses of *T. gondii* tachyzoites were required to induce BBB leakage similar to LPS. This strongly advocates against a generalized BBB dysfunction early in natural *T. gondii* infections, but does not exclude the occurrence of focalized dysregulations (*Estato et al., 2018*). In fact, focalized permeability elevations were detected in the vicinity of parasite foci, albeit in low numbers and without increased relative numbers over time. This is indicative of BBB permeability variations over time following the penetration of parasites to the parenchyma or that the BBB is transiently dysregulated. Thus, future investigations need to determine if an elevated BBB permeability around a given focus is consistently present, albeit transient, or if translocation can also occur in the absence of elevated permeability. Also, taking into consideration the alternative pathways of translocation discussed below, these two possibilities are not necessarily mutually exclusive. Noteworthy, despite the infection of endothelium and expected natural lysis of parasitized cells after replication, we found no evidence of irreversible vascular disruption with petechial hemorrhages, as described in cerebral

malaria (*Haldar et al., 2007*) and meningococcal disease (*Coureuil et al., 2014*). Yet, this does not rule out that single translocation events of *T. gondii* or egress of tachyzoites from single endothelial cells transiently destabilize the BBB below the detection threshold for the permeability tracer EB. In in vitro models using polarized primary brain endothelium, extracellular tachyzoites and parasitized dendritic cells rapidly translocated in absence of endothelial cell lysis, with a modest perturbation of the transcellular electrical resistance (TCER) but, importantly, in absence of leakage of a low-molecular weight permeability tracer (*Ross et al., 2019*; *Ross et al., 2021*). In sharp contrast, cellular depolarization events or lysis of cells in the monolayers following parasite replication at later time-points led to significant decreases in TCER and dramatic elevations of permeability tracer concentration in vitro (*Ross et al., 2019*; *Ross et al., 2021*). Altogether, these findings motivated an exploration of the BBB regulatory mechanisms in vivo.

We demonstrate a role for the BBB junction regulator FAK in the restricted passage of *T. gondii* to the brain parenchyma. Despite being embryonic lethal (*Ilić et al., 1995*), unchallenged adult mice with conditional deletion of endothelial FAK normally exhibit wildtype phenotype and apparently normal BBB morphology (*Lee et al., 2010*; *Weis et al., 2008*), consistent with the maintained morphology and the absence of leakage of blood tracer observed here. However, upon *T. gondii* challenge, FAK-deficient mice carried higher numbers of parasite foci and foci of larger size in the parenchyma, indicative of increased permissiveness compared to wildtype. Interestingly, pharmacological treatment with FAK inhibitor had a more pronounced effect on parasite loads with elevated BBB permeability, compared with conditional ablation of endothelial FAK. This difference may be due to a number of factors. First, conditional deletion of endothelial FAK is partially functionally compensated (*Weis et al., 2008*). Second, although the FAK inhibitor defactinib appears to be well tolerated, immunomodulatory effects may also contribute to the higher parasite loads in peripheral organs, and, thus, elevated numbers of parasites may enter the cerebral circulation (*Osipov et al., 2019*). Additionally, inhibition of FAK may interfere with parasite dissemination in infected leukocytes (*Ólafsson et al., 2019*; *Cook et al., 2018*) and parasite survival in endothelium (*Portillo et al., 2017*). Regardless, induced ablation of FAK in endothelium, with maintained expression of FAK in bone-marrow-derived cells, yielded higher numbers and size of parasite foci in brain parenchyma, demonstrating endothelium-related effects. Further, we recently reported that altered FAK phosphorylation upon *T. gondii* infection facilitates parasite passage across highly polarized primary brain endothelial cell monolayers (*Ross et al., 2019*). Importantly, this transient dysregulation of FAK occurs without disruption of monolayer polarization or increased permeability to low-molecular weight tracer, consistent with in vivo observations here. Thus, FAK inhibition or ablation appears to have a destabilizing, but non-disruptive, effect on the BBB. Jointly, the data show that endothelial FAK-regulated BBB restrictiveness is one of the determinants of *T. gondii* passage across the BBB.

Alternative routes have been proposed for the translocation of *T. gondii* across restrictive biological barriers (*Lambert and Barragan, 2010*), including the BBB (*Schlüter and Barragan, 2019*), and it is likely that more than one mechanism mediates entry into the CNS, including growth across endothelium, direct penetration and trafficking of leukocytes (*Matta et al., 2021*). The paracellular entry route (*Ross et al., 2019*; *Barragan et al., 2005*; *Barragan and Sibley, 2002*; *Furtado et al., 2012*; *Weight et al., 2015*) implies passing cellular TJs, and the transcellular route (*Konradt et al., 2016*; *Lambert and Barragan, 2010*; *Dubey, 1997*) implies apical parasite invasion and basolateral exit from the cells after replication. Additionally, leukocytes traffic into the brain parenchyma during toxoplasmosis (*John et al., 2011*) and parasitized hypermigratory leukocytes may aid in direct translocation (*Bhandage et al., 2020*; *Courret et al., 2006*; *Lachenmaier et al., 2011*) or by depositing parasites on the endothelium (*Lambert and Barragan, 2010*; *Baba et al., 2017*). Although the relative importance of the alternative pathways of passage is not specifically addressed here, the data are relevant for these alternatives or combinations of pathways because focal invasion events with an impact on TJ and BBB barrier integrity are likely to occur, for example, upon egress of tachyzoites from endothelium (*Konradt et al., 2016*). In these settings, we demonstrate that a non-replicating *T. gondii* line (CPS) transmigrates across polarized endothelium and epithelium in vitro, with elevated transmigration frequencies upon FAK inhibition. Thus, despite the reduced gliding motility and transmigration abilities of this mutant, the data provide additional evidence that FAK impacts the endothelial process of tachyzoite transmigration, in absence of monolayer disruption and independently of parasite replication in vitro. In mice, surprisingly few CPS parasites were detected in the brain and other peripheral

organs shortly after inoculation, consistent with the absence of symptomatology in mice despite the inoculation of doses that are imminently lethal with wildtype ( $> 10^8$ ) and with scarce localization of single CPS tachyzoites intravascularly, and extravascularly in the brain parenchyma. Partly contrasting with in vitro results, elevated total numbers of CPS parasites were not detected in the brain upon FAK inhibition or ablation in vivo. Given the high TCER of the BBB, this may be related to the expected overall reduced invasive and transmigration abilities of this mutant, in combination with a sensitivity limitation of the assay due to scarce total invasion events. Importantly, the low numbers of intravascular CPS parasites compared with wildtype (RH) precluded direct comparisons with wildtype for numbers of translocation events in vivo. Together, the reduced gliding and TCER-related transmigration abilities of this auxotroph mutant upon uracil deprivation indicate that metabolic stress (*Blondy et al., 2020*) negatively impacts invasion events in vitro and in vivo and, that this mutant is rapidly neutralized in vivo, possibly as a consequence of its generally reduced invasion capability.

Jointly, the present study identifies FAK-regulated TJ stability as a determinant of *T. gondii* translocation across cortical capillaries. We postulate that the infection-related destabilization of the BBB by FAK dephosphorylation (*Ross et al., 2019*) facilitates active transmigration of parasites across the BBB. The impact of transient and focal BBB dysregulation on the alternative pathways of parasite passage to the parenchyma, together with the roles of the non-endothelial cellular constituents of the neurovascular unit, need to be addressed in future investigations.

In humans and mice, primary infection with *T. gondii* entails invasion of the CNS with passage of the parasite across the BBB. Natural infections in humans normally present mild or no symptomatology (*Montoya and Liesenfeld, 2004*). Thus, our data supports the concept that passage of *T. gondii* to the brain parenchyma is an early and clinically *silent* process in absence of focal neurological symptoms or cardinal signs of inflammation. As the immune response develops, tachyzoites are neutralized and developmental switch to bradyzoite stages takes place for chronic perseverance in the brain parenchyma.

# Materials and methods

**Key resources table**

| Reagent type (species) or resource | Designation | Source or reference | Identifiers | Additional information |
|---|---|---|---|---|
| Strain, strain background (*Mus musculus*) | C57BL/6NCrl (B6) | Charles River Laboratories | Strain code 027 | |
| Strain, strain background (*Mus musculus*) | Crl:CD1(ICR) (CD1) | Charles River Laboratories | Strain code 022 | |
| Strain, strain background (*Mus musculus*) | BALB/cAnNCrl (BALB/c) | Charles River Laboratories | Strain code 028 | |
| Strain, strain background (*Mus musculus*) | B6.129P2(FVB)-Ptk2tm1.1Guan/J (*Ptk2*flox/flox) | Jackson Laboratories (*Shen et al., 2005*; *Daneman and Prat, 2015*) | JAX stock # 031956 | |
| Strain, strain background (*Mus musculus*) | C57BL/6-Tg(Cdh5-cre/ERT2)1Rha (*Cdh5*cre+/-) | Taconic Biosciences | Taconic Model # 13,073 | |
| Cell line (*T. gondii*) | GFP-expressing RH-LDM, type I (RH) | *Kim et al., 2001*; *Barragan and Sibley, 2002*; | | |
| Cell line (*T. gondii*) | GFP-expressing ME49/PTG, type II (ME49) | *Kim et al., 2001* | | |
| Cell line (*T. gondii*) | mCherry-expressing CPS, type I (CPS) | *Fox and Bzik, 2002*; *Konradt et al., 2016* | | |

*Continued on next page*

*Continued*

| Reagent type (species) or resource | Designation | Source or reference | Identifiers | Additional information |
|---|---|---|---|---|
| Cell line (*Homo sapiens*) | HFF-1 (HFF) | American Type Culture Collection | SCRC-1041 | |
| Cell line (*Mus musculus*) | bEND.3 | American Type Culture Collection | CRL-2299 | |
| Cell line (*Homo sapiens*) | Caco-2 | American Type Culture Collection | HTB-37 | |
| Antibody | Anti-P-glycoprotein (Mouse monoclonal) | GeneTex | Cat# GTX23364, RRID:AB_367204 | IF(1:100) |
| Antibody | Anti- GFAP (Rat monoclonal) | Thermo Fisher Scientific | Cat# 13–0300, RRID:AB_2532994 | IF(1:200) |
| Antibody | Anti-Occludin (Mouse monoclonal) | Thermo Fisher Scientific | Cat# 33–1500, RRID:AB_2533101 | IF(1:100) |
| Antibody | Anti- Iba1(Rabbit polyclonal) | FUJIFILM Wako Shibayag | Cat# 019–19741, RRID:AB_839504 | IF(1:100) |
| Antibody | Anti-CD45 (Rat monoclonal) | BD Biosciences | Cat# 550539, RRID:AB_2174426 | IF(1:20) |
| Antibody | Anti-CD31 (Rat monoclonal) | BD Biosciences | Cat# 553370, RRID:AB_394816 | IF(1:100) |
| Antibody | Anti-ZO-1 (Rabbit polyclonal) | Thermo Fisher Scientific | Cat# 40–2200, RRID:AB_2533456 | IF(1:100) |
| Antibody | Anti-Glut1(Goat polyclonal) | Santa Cruz Biotechnology | Cat# sc-1605, RRID:AB_2239463 | IF(1:40) |
| Antibody | Anti- laminin α4 (Rat polyclonal) | gift from Lydia Sorokin and Krister Kristensson.*Masocha et al., 2004*. doi:10.1172/JCI22104. | serum 341 | IF(1:500) |
| Antibody | Anti- laminin α5 (Rat polyclonal) | gift from Lydia Sorokin and Krister Kristensson. *Masocha et al., 2004*. doi:10.1172/JCI22104. | supernatant; clone 4G6 | IF(1:20) |
| Antibody | Anti- laminin β1 (Rat monoclonal) | gift from Lydia Sorokin and Krister Kristensson. *Masocha et al., 2004*. doi:10.1172/JCI22104. | 1:1; clone 3A4 | IF(1:100) |
| Antibody | Anti-*T. gondii* (Rabbit polyclonal) | *Dellacasa-Lindberg et al., 2011*:10.1128/IAI.01042–10 | | IF(1:100) |
| Antibody | Anti-Rat IgG (H + L) Alexa Fluor 594 (Chicken polyclonal) | Thermo Fisher Scientific | Cat# A-21471, RRID:AB_2535874 | IF(1:500) |
| Antibody | Anti-Mouse IgG (H + L) Alexa Fluor 405 (Goat polyclonal) | Thermo Fisher Scientific | Cat# A-31553, RRID:AB_221604 | IF(1:500) |
| Antibody | Anti-Rabbit IgG (H + L) Alexa Fluor 488 (Chicken polyclonal) | Thermo Fisher Scientific | Cat# A-21441, RRID:AB_2535859 | IF(1:1000) |

*Continued on next page*

*Continued*

| Reagent type (species) or resource | Designation | Source or reference | Identifiers | Additional information |
|---|---|---|---|---|
| Antibody | Anti-Rabbit IgG (H + L) Alexa Fluor 594 (Chicken polyclonal) | Thermo Fisher Scientific | Cat# A-21442, RRID:AB_2535860 | IF(1:1000) |
| Antibody | Anti-Goat IgG (H + L) Alexa Fluor 488 (Donkey polyclonal) | Jackson ImmunoResearch Labs | Cat# 705-545-147, RRID:AB_2336933 | IF(1:100) |
| Antibody | Anti-Rat IgG (H + L) Alexa Fluor 488 (Donkey polyclonal) | Jackson ImmunoResearch Labs | Cat# 712-545-150, RRID:AB_2340683 | IF(1:100) |
| Antibody | Anti-mouse CD45 Brilliant Violet 711(Rat monoclonal) | BioLegend | Cat# 103147, RRID:AB_2564383 | FACS (1 ul per test) |
| Antibody | Anti-mouse CD11b (M1/70), APC, eBioscience (Rat monoclonal) | Thermo Fisher Scientific | Cat# 17-0112-82, RRID:AB_469343 | FACS (1 ul per test) |
| Antibody | Anti- FAK (Rabbit polyclonal) | Cell Signaling Technology | Cat# 3285, RRID:AB_2269034 | WB (1:1000) |
| Antibody | Anti-Toxoplasma gondii P30 (Mouse monoclonal) | Thermo Fisher Scientific | Cat# MA1-83499, RRID:AB_935764 | IF(1:50) |
| Antibody | Anti-rabbit IgG, HRP-linked (Goat polyclonal) | Cell Signaling Technology | Cat# 7074, RRID:AB_2099233 | WB (1:3000) |
| Antibody | Anti-GAPDH (Rabbit polyclonal) | Millipore | Cat# ABS16, RRID:AB_10806772 | WB (1:3000) |
| Sequence-based reagent | Gapdh_F | This paper | PCR primers | TGACCTCAA CTACATGGTCTACA |
| Sequence-based reagent | Gapdh_R | This paper | PCR primers | CTTCCCATT CTCGGCCTTG |
| Sequence-based reagent | Hprt_F | This paper | PCR primers | CCC AGC GTC GTG ATT AGC |
| Sequence-based reagent | Hprt_R | This paper | PCR primers | GGA ATA AAC ACT TTT TCC AAA TCC |
| Sequence-based reagent | Mmp2_F | This paper | PCR primers | GTT GCT TTT GTA TGC CCT TCG |
| Sequence-based reagent | Mmp2_R | This paper | PCR primers | TCA GAC AAC CCG AGT CCT TTG |
| Sequence-based reagent | Mmp12_F | This paper | PCR primers | TGT GGA GTG CCC GAT GTA CA |
| Sequence-based reagent | Mmp12_R | This paper | PCR primers | AGT GAG GTA CCG CTT CAT CCA T |
| Sequence-based reagent | Mmp9_F | This paper | PCR primers | AAA ACC TCC AAC CTC ACG GA |
| Sequence-based reagent | Mmp9_R | This paper | PCR primers | GCT TCT CTC CCA TCA TCT GGG |
| Sequence-based reagent | Mmp14_F | This paper | PCR primers | TATGGGCCC AACATCTGTGAC |
| Sequence-based reagent | Mmp14_R | This paper | PCR primers | AACCATC GCTCCTTG AAGACA |

*Continued on next page*

Continued

| Reagent type (species) or resource | Designation | Source or reference | Identifiers | Additional information |
|---|---|---|---|---|
| Sequence-based reagent | Vcam1_F | This paper | PCR primers | GTG ACT CCA TGG CCC TCA CTT |
| Sequence-based reagent | Vcam1_R | This paper | PCR primers | CGT CCT CAC CTT CGC GTT TA |
| Sequence-based reagent | Icam1_F | This paper | PCR primers | CAA TTT CTC ATG CCG CAC AG |
| Sequence-based reagent | Icam1_R | This paper | PCR primers | CTG GAA GAT CGA AAG TCC GG |
| Sequence-based reagent | Sele (E-selectin)_F | This paper | PCR primers | CCC TGC CCA CGG TAT CAG |
| Sequence-based reagent | Sele (E-selectin)_R | This paper | PCR primers | ACG TGC ATG TCG TGT TCCA |
| Sequence-based reagent | Ocln_F | This paper | PCR primers | AGG ACG GAC CCT GAC CAC TA |
| Sequence-based reagent | Ocln_R | This paper | PCR primers | GGT GGA TAT TCC CTG ACC CAG |
| Sequence-based reagent | Ifng_F | This paper | PCR primers | GCT TTG CAG CTC TTC CTC AT |
| Sequence-based reagent | Ifng_R | This paper | PCR primers | CAC ATC TAT GCC ACT TGA GTT AAA ATA GT |
| Sequence-based reagent | Ccl2_F | This paper | PCR primers | CATCCACGTGTTGGCTCA |
| Sequence-based reagent | Ccl2_R | This paper | PCR primers | GATCATCTTGCTGGTGAATGAGT |
| Sequence-based reagent | Il1240_F | This paper | PCR primers | TCCCTCAAGTTCTTTGTTCG |
| Sequence-based reagent | Il1240_R | This paper | PCR primers | CGCACCTTTCTGGTTACAC |
| Sequence-based reagent | Timp1_F | This paper | PCR primers | GCAACTCGGACCTGGTCATAA |
| Sequence-based reagent | Timp1_R | This paper | PCR primers | CGCTGGTATAAGGTGGTCTCG |
| Sequence-based reagent | Timp2_F | This paper | PCR primers | CTCGCTGTCCCATGATCCC |
| Sequence-based reagent | Timp2_R | This paper | PCR primers | GCCCATTGATGCTCTTCTCTGT |
| Sequence-based reagent | B1_F | This paper | PCR primers | GCATTGCCCGTCCAAACT |
| Sequence-based reagent | B1_R | This paper | PCR primers | AGACTGTACGGAATGGAGACGAA |
| Sequence-based reagent | Ptk2tm1.1_F | The Jackson Laboratory | Stock No: 031956. Protocol 35,084 | GAA CTT GAC AGG GCT GGT CT |
| Sequence-based reagent | Ptk2tm1.1_R | The Jackson Laboratory | Stock No: 031956. Protocol 35,084 | CTC CAG TCG TTA TGG GAA ATC T |
| Sequence-based reagent | CRE_F | Taconic Biosciences | A.2684.Line: 13,073 Tg(Cdh5-cre/ERT2)1Rha | GCCTGCATTACCGGTCGATGCAACGA |

*Continued*

| Reagent type (species) or resource | Designation | Source or reference | Identifiers | Additional information |
|---|---|---|---|---|
| Sequence-based reagent | CRE_R | Taconic Biosciences | A.2684.Line: 13,073 Tg(Cdh5-cre/ERT2)1Rha | GTGGCAGATGGCGCGGCAACACCATT |
| Chemical compound, drug | Lipopolysaccharides from *Escherichia coli* O111:B4 | Sigma Aldrich | L2630 | |
| Chemical compound, drug | Hydrocortisone | Sigma Aldrich | H0888 | |
| Chemical compound, drug | Evans Blue | Sigma Aldrich | E2129 | |
| Chemical compound, druC | Defactinib (VS-6063) | Selleck Chemical's | S7654 | |
| Chemical compound, drug | (Z)–4-Hydroxytamoxifen | Sigma Aldrich | H7904 | |
| Chemical compound, drug | Tamoxifen | Sigma Aldrich | T5648 | |
| Commercial assay or kit | RNeasy Mini Kit | QIAGEN | 74,104 | |
| Commercial assay or kit | DNeasy Blood & Tissue Kit | QIAGEN | 69,506 | |
| Commercial assay or kit | SuperScript IV Reverse Transcriptase (RT) | Thermo Fisher Scientific | 18090050 | |
| Commercial assay or kit | Platinum Taq DNA Polymerase | Thermo Fisher Scientific | 10966034 | |

## Experimental animals

All utilized mice strains are listed in the Key Resources Table. B6.129P2(FVB)-*Ptk2*[tm1.1Guan]/J (*Ptk2*[flox/flox]) mice were crossed with C57BL/6-Tg(Cdh5-cre/ERT2)1Rha (*Cdh5*[cre+/-]) mice for endothelial cell-specific deletion of FAK-encoding *Ptk2*. Experimental mice were *Ptk2*[flox/flox]*Cdh5*[cre+/-] (ECPTK2 cKO) and *Ptk2*[flox/flox]*Cdh5*[cre-/-] (ECPTK2 WT) littermates were used as control wild-type mice. *Ptk2*[wt/wt]*Cdh5*[cre-/-] and *Ptk2*[wt/wt]*Cdh5*[cre+/-] mice were also used in control experiments. Mice were injected intraperitoneally (i.p.) with 2 mg tamoxifen (Sigma-Aldrich) for 5 consecutive days to induce recombination. Six- to 12-week-old male or female mice were used for experiments. Animals were housed under specific pathogen-free conditions at the Experimental Core Facility, Stockholm University.

## Parasite and cell lines

*T. gondii* tachyzoites were maintained by serial 2 day passage in human foreskin fibroblast (HFF) monolayers cultured in Dulbecco's modified Eagle's medium (DMEM; Thermofisher scientific) with 10% fetal bovine serum (FBS; Sigma), gentamicin (20 µg/ml; Gibco), glutamine (2 mM; Gibco), and HEPES (0.01 M; Gibco). For the CPS line, uracil (250 µM; Sigma) was added to the culture medium as indicated (*Fox and Bzik, 2002*). Cultures were periodically tested for mycoplasma and tested negative. Parasite lines and cell lines are listed in the Key Resources Table.

## Infections and treatments in mice

Mice were inoculated intravenously (i.v.) with high doses (hd, $1–10 \times 10^6$) or i.p. with low doses (ld, $2,5–5 \times 10^4$) of freshly egressed *T. gondii* tachyzoites. To evaluate the effect of inflammation on parasite loads, *T. gondii*-infected CD1 mice were inoculated i.p. with LPS (serotype 011:B4, Sigma) at 0.08 or 0.2 mg/kg/day for seven or 5 days, respectively, starting 1 day before infection or treated subcutaneously with hydrocortisone (Cortisol, Sigma-Aldrich) at 20 or 40 mg/kg/day for 6 days, starting 3 days before *T. gondii* infection. For assessment of BBB permeability, control mice were injected i.p. with LPS (2 mg/kg) and sacrificed 6 h post-inoculation. *T. gondii*-infected mice were inoculated i.p. with

the FAK inhibitor VS-6063 (defactinib; Selleckchem) at 50 mg/kg twice a day, starting 1 day before *T. gondii* infection.

## Immunohistochemistry

To visualize the lumen of cerebral blood vessels, mice were injected i.v. with 200 µl of Evans blue solution (EB) containing 3% Evans blue fluorescent dye (Sigma) and 5% BSA dissolved in phosphate-buffered saline (PBS), prior to organ extraction. Extracted organs were fixed in 4% paraformaldehyde 0,1 M phosphate buffer (PFA) for 2 days and preserved in sucrose solution (10% sucrose, 0.01% NaN3 and 0.1 M Sorensen's phosphate buffer) for 4 days. Tissues were then frozen on dry ice and 14–50 µm cryosections collected onto gelatin-coated slides. For immunostaining, brains were dissected, snap frozen on dry ice and stored in –80 ° C until sectioned in a cryostat. Brain cryosections of 14 µm thickness on coronal plane were fixed in 4% PFA followed by acetone fixation, permeabilization (0.2% Saponin or 0.3% Triton X-100 in PBS), blocking (1% BSA, 1%–10% FBS) and incubation with primary antibodies in IHC buffer (1% BSA, 0.3% Triton X-100, 0.01% NaN3) ON at 4 °C or for 3 hr at RT. All sections were then washed in PBS and incubated with corresponding secondary antibodies diluted in IHC buffer for 1 hr at RT, followed by DAPI staining. Microvessels were purified as describe below and seeded onto glass coverslips pre-coated with 1% gelatin, fixed in 4% PFA for 5 min, rinsed in PBS, incubated ON at 4° C with primary antibodies diluted in IHC buffer, and finally stained with secondary antibodies, followed by DAPI staining. All primary and secondary antibodies used are listed in Key Resources Table.

## Microscopy and image analysis

Epifluorescence microscopy was performed by using a 20 X objective (Observer Z1, Zeiss) or 63 X objective (DMi8, Leica Microsystems). Z-stack images were captured by confocal microscopy (LSM 800 Airyscan, Zeiss) and processed with ZenBlue software (Zeiss). Two-photon imaging was performed using a Leica SP8 DIVE microscope equipped with 4Tune non-descanned spectral detectors, a HC PL IRAPO 40 x/1.10 water immersion objective (Leica Microsystems) and a tunable INSIGHT DUAL X3 laser (Spectra-Physics, US). GFP was excited with 924 nm laser pulses, emission was detected at 500–520 nm, Evans blue was excited at 945 nm and detected at 620–750 nm. 3D reconstruction of z-stacks was performed using the 3D Visualization function in the Leica Application Suite X (LAS X, Leica Microsystems). Hyvolution, a Huygens-powered (Scientific Volume Imaging B.V.) deconvolution function integrated in LAS-X was used for deconvolution. To analyze distances between replicating tachyzoites and vasculature, confocal images were processed with 3D view using Imaris9.5 software. The brain microvasculature was defined by using the *Surface* rendering tool and replicating tachyzoites were identified as minimum 2 µm spots. The distance from each spot to the border of all surfaces was calculated using the *ShortestDistanceToSurfaces* tool.

## Characterizations and quantifications of parasites in the CNS

Quantifications of *T. gondii* (GFP$^+$) foci were expressed as number of foci/mm$^2$ of tissue or per section on coronal brain sections from parenchyma, choroid plexus and meninges. A parasite focus was defined as a group of vacuoles with replicating *T. gondii* (GFP$^+$). Unless differently stated, individual vacuoles with replicating *T. gondii* tachyzoites (two or more tachyzoites per vacuole) were counted. The size of the foci was determined by counting number of vacuoles and categorized as follows <10, 11–50 or >50 vacuoles/focus. Single tachyzoites were also counted in *Figures 3e, h–k , and 9*. Detailed localization analyses of *T. gondii* in relation to the cerebral vasculature were performed in the prefrontal cortex area of the brain parenchyma (bregma +2.96 mm to bregma +1.70 mm), using epifluorescence and confocal microscopy (63 X objective). Parasite (GFP$^+$) signal co-localizing with EB signal was defined as intravascular. Parasite signal localized ≤10 µm and >10 µm from the EB signal of the vascular lumen was defined as perivascular and parenchymal, respectively. The vascular lumen and diameter sizes (µm) were determined by a scale-calibrated measurement (ImageJ software) across the EB signal and α5 laminin signal, respectively, in areas with GFP$^+$ foci. To calculate the α5 laminin/P-glycoprotein fluorescence ratio, the mean α5 laminin fluorescence intensity signal was divided by the mean P-glycoprotein fluorescence intensity signal for each vessel. Plaquing assays were performed as described (*Fuks et al., 2012*). Briefly, brains, livers, and spleens were extracted and homogenized

on 70 µm cell strainers. The numbers of viable parasites per g of tissue were determined by plaque formation on HFF-1 monolayers.

## Assessment of BBB permeability

To assess BBB permeability, mice were injected i.v. with 200 µl of EB solution 15–20 min prior to organ extraction. Brains were dissected and processed for immunohistochemistry as described above. Pixel intensities of EB signal from the vessels and surrounding parenchyma were analyzed by ImageJ software as detailed in *Figure 5—figure supplement 1*. On average 75 blood vessels per mouse from a minimum of 5 fields of view (magnification 20 x) per brain section were analyzed. All images were acquired using identical microscope settings.

## Isolation and treatments of brain microvessels and murine brain endothelial cells (MBECs)

Cerebral microvessels from infected and non-infected mice were isolated as described (*Ross et al., 2019*; *Howland et al., 2015*). Briefly, brains were minced with a scalpel-blade and homogenized by passing through a 23-gauge needle. The brain homogenate was mixed in an equal volume of 30% dextran solution (MW 70,000; Sigma-Aldrich) and centrifuged at 10,000 g for 15 min at 4° C. After removal of myelin (top layer), the pellet was resuspended in PBS and passed through a 40 µm cell strainer. The cell strainer was then back-flushed with PBS to collect the microvessel fragments. For in vitro culture of MBECs, microvessels from non-infected mice were incubated with 1 mg/ml Collagenase IV (Gibco) and 200 U/ml of DNAse I (Roche) dissolved in HBSS ($Ca^{2+}$ and $Mg^{2+}$) for 1.5–2 hr at 37 ° C. Digested microvessels were washed and seeded in 12-well plates pre-coated with 1% gelatin (Gibco) in EBM-2/EGM-2 medium (Lonza) with 4 µg/ml puromycin (Santa Cruz). The purity of MBECs was determined by CD31 expression (70%–80%) and ZO-1 expression as detailed in *Ross et al., 2019*. MBECs were challenged with freshly egressed *T. gondii* tachyzoites (RH-LDM, MOI 4) or LPS (100 ng/ml), for six or 12 h, and processed for PCR.

## Gliding and transmigration assays

Gliding assays were performed as described (*Barragan and Sibley, 2002*). Briefly, glass slide coverslips were coated with FBS. Recently egressed tachyzoites were added to the coverslips and incubated at 37 ° C for 1 hr before fixation in 3.7% formalin-PBS. Detection of trails and length analyses were performed by using SAG1 mAb (P30) directly conjugated to Alexa 488 fluorochrome, epifluorescence microscopy and ImageJ software. Random trails associated to a tachyzoite were counted and tachyzoites with absence of trail were not counted.

Transmigration assays were performed as described (*Ross et al., 2019*). Briefly, freshly egressed tachyzoites ($2 \times 10^5$) were exposed to polarized bEnd.3 and Caco2 monolayers in transwells. Polarization and permeability parameters were determined by measurement of transcellular electrical resistance (TCER) and FITC-dextran (3 kDa), before and after the assay, as detailed in *Ross et al., 2019*. Following incubation overnight (16–18 hr), transmigrated tachyzoites were quantified by plaquing assays, with uracil added to the medium for the CPS line. Cell lines and reagents are listed in Key Resources Table.

## Tissue preparation and flow cytometry

To prepare single-cell suspensions, brain tissue was incubated with 1 mg/ml collagenase IV (Gibco) and 10 µg/ml DNase I (Roche) for 1 hr at 37 °C. Brain homogenates were then washed with FACS buffer (0.5% FBS, 2 mM EDTA in PBS), centrifuged at 250 g for 10 min and pellets resuspended in Percoll (Easycoll, Biochrom) in PBS at a final concentration of 30%. After centrifugation at 500 g for 30 min, the myelin (top layer) was discarded. Cell pellets were filtered through a 40 µm cell strainer and resuspended in FACS buffer. Spleen and lung cells were isolated by mechanical homogenization on 70 µm cell strainers followed by treatment with red blood cell lysis buffer (0.802% NH4Cl, 0.08%NaHCO3, 0.037% EDTA). Freshly isolated cells from brains, spleens, and lungs were blocked for 30 min in Fc block (BD Biosciences), and incubated with antibodies against CD45 (BV711, BioLegend) and CD11b (APC, eBioscience). Signal from GFP-expressing *T. gondii* was detected in the filter 525/550 channel. Flow cytometry was performed on LSR Fortessa (BD Biosciences) and data were analyzed with FlowJo software v10.

## Real-time quantitative PCR

Total RNA was extracted from brain, liver or spleen homogenates using TRI Reagent (Sigma), chloroform and isopropanol. Microvessels and MBEC RNA was extracted using the RNeasy Mini kit (Qiagen) according to the manufacturer's protocols. The amount and purity of isolated RNA was determine using a spectrophotometer (NanoDrop 1000, ThermoFisher Scientific) by reading the absorbance at 260 nm and 280 nm. The ratio A260/A280 of RNA from homogenate brain samples was in the range of 1.91–2.06, for microvessels 1.8–2.0 and for MBEC 2.03–2.16. First-strand cDNA was synthesized with Superscript IV Reverse Transcriptase (Invitrogen). Real-time PCR was performed using SYBR green PCR master mix (Kapa Biosystems), forward and reverse primers (200 nM) and cDNA (0.5–1 μg) with a QuantStudio TM 5 Real-Time PCR system (Thermofisher). ΔCt values were calculated with the house-keeping genes HPRT and GAPDH. $2^{-\Delta\Delta Ct}$ values were used to calculate fold-change of expression upon challenge in cells and tissues. All primers were designed using Get-prime or Primer-BLAST software (Key Resources Table).

## Western blot

To quantify protein levels of endothelial FAK in *Cdh5*<sup>*cre+/-*</sup> (*Ptk2*<sup>*wt/wt*</sup>, *Ptk2*<sup>*flox/wt*</sup> or *Ptk2*<sup>*flox/flox*</sup>) mice upon tamoxifen treatment, MBECs were isolated and treated with (Z)–4-Hydroxytamoxifen in vitro (500 nM; Sigma) for 24 hr on day 10. Bone-marrow-derived cells were isolated from mice as described (*Bhandage et al., 2020*). Livers were extracted and homogenized. MBECs (day 14), BMDCs (day 7) or liver extracts were lysed in RIPA buffer (150 mM NaCl, 50 mM Tris, 0.1% Triton, 0.5% deoxycholic acid, 0.1% SDS) with cOmplete mini protease and phosphatase inhibitors (Roche). Proteins were separated using 8% SDS-PAGE gels, blotted onto a PVDF membrane (Millipore) and blocked (5% BSA, Sigma-Aldrich) followed by incubation with anti-FAK or anti-GAPDH followed by anti-rabbit HRP. Proteins were detected by mean of enhanced chemiluminescence (GE Healthcare) in a BioRad ChemiDoc XRS<sup>+</sup>. Densitometry analyses was performed using ImageJ (NIH, MD, USA).

## Genotyping

Mouse genomic DNA was isolated from ear biopsies using DNeasy Blood & Tissue kit (Qiagen). PCR was performed as described by Taconic Biosciences for *Cdh5*<sup>*cre+/-*</sup> mice, and as described by Jackson Laboratory for *Ptk2*<sup>*flox/flox*</sup> mice.

## Statistical analyses

Statistical analyses were performed using GraphPad Prism software (v. 8) or TIBCO Statistica software (v. 7). For two-group comparisons, two-tailed Student's t-test was performed on samples with normal distribution and Mann-Whitney test was used on non-normally distributed data. For multiple comparisons, data with a normal distribution were analyzed by one-way ANOVA, followed by Tukey's test or Dunnett's test. ANOVA and ANCOVA- F value and degrees of freedom F (dfn, dfd) are indicated in the figure legends. For multiple comparisons, Kruskal-Wallis test, followed by Dunn's post-hoc test was performed when data was not normally distributed. For all statistical tests, values of $p \geq 0.05$ were defined as non-significant, and $p < 0.05$ were defined as significant.

## Acknowledgements

We acknowledge the intravital imaging facility at Stockholm university (IVMSU) and the National Microscopy Infrastructure (NMI), Drs. Lydia Sorokin and Krister Kristensson for providing laminin antibodies and Drs. Christopher Hunter and David Bzik for uracil auxotroph parasites. This work was funded by the Swedish Research Council (Vetenskapsrådet, 2018–02411) and the Olle Engkvist Foundation (193-609).

## Additional information

### Funding

| Funder | Grant reference number | Author |
|---|---|---|
| Vetenskapsrådet | 2018-02411 | Antonio Barragan |
| Olle Engkvist Foundation | 193-609 | Antonio Barragan |

The funders had no role in study design, data collection and interpretation, or the decision to submit the work for publication.

### Author contributions

Gabriela C Olivera, Data curation, Formal analysis, Investigation, Methodology, Visualization, Writing – original draft, Writing – review and editing; Emily C Ross, Data curation, Formal analysis, Investigation, Methodology, Validation, Visualization, Writing – original draft, Writing – review and editing; Christiane Peuckert, Data curation, Formal analysis, Methodology, Validation, Visualization; Antonio Barragan, Conceptualization, Data curation, Formal analysis, Funding acquisition, Investigation, Project administration, Resources, Supervision, Writing – original draft, Writing – review and editing

### Author ORCIDs

Antonio Barragan ⬦ http://orcid.org/0000-0001-7746-9964

### Ethics

All animal experimentation was approved by the Regional Animal Research Ethical Board, Stockholm, Sweden, (protocol numbers N135/15, 9707-2018 and 14458-2019), following proceedings described in EU legislation (Council Directive 2010/63/EU).

### Decision letter and Author response

Decision letter https://doi.org/10.7554/eLife.69182.sa1
Author response https://doi.org/10.7554/eLife.69182.sa2

## Additional files

### Supplementary files

• Transparent reporting form

### Data availability

All data generated or analysed during this study are included in the manuscript and supporting files.

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
