## [Editor Report]

There is significant interest in how pathogens like Toxoplasma infect the brain. The present study carefully details early steps in this process by inoculating mice intravenously and monitoring their distribution within brain tissues over the subsequent days. The analysis presented will be a valuable reference for further investigation of this process. The authors perform several perturbations that involve the permeability of the blood-brain barrier as a key determinant of parasite entry into the brain parenchyma. Taken together the study highlights the critical role of the blood-brain barrier in restricting access of Toxoplasma to the brain.

---

## [Decision Letter]

**Decision letter after peer review:**

Thank you for submitting your article "Blood-brain barrier-restricted translocation of *Toxoplasma gondii* from cortical capillaries" for consideration by *eLife*. Your article has been reviewed by 3 peer reviewers, one of whom is a member of our Board of Reviewing Editors, and the evaluation has been overseen by Dominique Soldati-Favre as the Senior Editor. The reviewers have opted to remain anonymous.

Essential revisions:

Detailed comments from reviewers are copied below for your reference. The reviewers were enthusiastic about the potential of the study; however, there was a strong sense that the main hypothesis and title of the study should directly tested as detailed in point 1 below. The reviewers felt that any outcome would be complementary to the present results and should be reported alongside them, either validating the hypothesis or modifying the conclusions of the manuscript.

1. Request for additional data. The authors should provide direct evidence for the hypothesis that FAK inhibition or deletion in the vascular endothelium alters parasite translocation by specifically altering barrier function. The reviewers note the various confounding effects of FAK disruption, which extend beyond its role in regulating BBB intercellular junctions. The reviewers were unanimous in requiring experimental evidence using non-replicating strains of Toxoplasma (like CPS) to test this hypothesis since, based on the current model, intravenous injection into WT and BBB FAK KO mice would result in increased translocation of the non-invasive parasites in the latter.

2. Request for textual modifications or clarification. The itemized reviews list several requests for clarification that can be addressed through changes to the text and the figures. Please respond to all such requests in the resubmission.*Reviewer #1:*

The critically sensitive nature of the vertebrate brain requires specific adaptations in barrier function and immune response, which ultimately help prevent damage associated with infections. Certain pathogens, like *Toxoplasma gondii*, can nevertheless translocate into the brain, so the mechanisms they use to achieve this are of clinical importance. The present study details the early steps of this process by inoculating mice intravenously and monitoring the distribution of parasites within brain tissues over the subsequent days. The result is a highly detailed picture of the early stages of brain infection that recapitulates previous results, and extends their conclusions by examining the role of the blood-brain-barrier (BBB) in this process. Inflammatory stimuli like LPS, which increase BBB permeability, also increased the penetration of Toxoplasma into the parenchyma. Inhibition or knockout of the focal adhesion kinase (FAK), which is important for the maintenance of the endothelial barrier, similarly increased the number of parasites found in the parenchyma; however, the interpretation of these observations is severely compromised by the other functions of FAK, including some within immune cells that control parasite function.

The authors have generated a detailed and quantitative picture of brain tissue infection following intravenous inoculation with *Toxoplasma gondii*. The results align well with findings from Konrad et al., (2016) using a similar model. Importantly, the authors examine how entry of parasites into the parenchyma is affected by a variety of perturbations. While many of the perturbations used ultimately affect parasite translocation into the brain, they lack the precision to make specific mechanistic conclusions. Moreover, the authors fail to dissect the contribution of parasite replication within endothelial cells, which would allow them to firmly establish the contribution of the blood-brain barrier in restricting parasite translocation into the brain.

1. It seems critical to determine what would be expected from randomly localized vacuoles. E.g. In Figure 3, what would be the average distance of random points within the imaged section from a capillary or a vascular branching point, and what would be the diameter of randomly selected capillaries within a field. Only then can we know whether the observed distribution is unusual.

2. Can the authors firmly conclude that at low parasite levels there is no permeability? The authors should consider whether single translocation events would provide sufficient signal to detect vascular leakage.

3. The authors need to exclude replication within endothelial cells to firmly establish that the BBB restricts parasite translocation. This could be accomplished by inoculating parasites incable of replication in mice. Without this evidence, it is formally possible that FAK inhibition/knockout or LPS are simply affecting parasite replication within endothelial cells.

4. Related to point 3, the number of parasite foci trends up in both perturbations of FAK (inhibition and knockout), within the intravascular compartment. Are investigators truly capable of distinguishing between perivascular and intravascular infection, given the lack of a specific marker for endothelial cells? The concern is that in some cases leakage of the endothelium will land parasites in perivascular sites that would look microscopically very similar to vascular sites; this effect could explain the surge in intravascular foci for perturbations that should alter permeability.*Reviewer #2:*

This paper examines how *T. gondii* parasites enter the brain, and the factors that limit parasite entrance. Similar to an earlier study (Konradt et al., 2016) the authors use intravenous inoculation of mice with fluorescent *T. gondii* tachyzoites, and microscopy to characterize the early events by which parasites enter the brain. As in Konradt et al., the current study provides evidence that parasites are first found in brain capillaries, and then replicate in cerebral endothelial cells prior to infecting the brain parenchyma. Interestingly, parasite replication in cerebral endothelial cells and entry into brain parenchyma observed at days 3-5 post infection does not immediately trigger permeability of the BBB, nor does it lead to substantial leukocyte infiltration. Rather, the increase in BBB permeability (as measured by an increase in fluorescence with vascular tracer Evans Blue) is first observed at day 10 post infection, coinciding with leukocyte infiltration. The authors utilize several imaging strategies to investigate the dynamics of *T. gondii* entry into the brain. The images produced are compelling and support the conclusions made by the authors, and in general the analysis being performed is clear and compelling. Additionally, the techniques used to measure the presence of inflammation or to manipulate the inflammatory environment are logical and produce compelling results. One area of concern is the potential impact of the non-physiological infection route (high dose intravenous tachyzoites) and the use of a highly virulent, non-cyst forming strain of the parasite (RH) which may lead to altered entry routes to those that occur during natural infection.

1. There is some concern that i.v. injection of large numbers of free tachyzoites might not accurately recapitulate events during normal infection. In addition the majority of experiments used the highly virulent non-cyst forming strain of the parasite (RH), which may use a different route of entry compared to less virulent, cyst forming strains. The authors should acknowledge and address these caveats in the text.

2. The conclusion that parasite entry (days 3-5) precedes BBB permeability, leukocyte entry and inflammation (day 10) is drawn from comparison of data from Figures 4, 5 and 6. However, the experiments in these 3 figures uses a variety of different infection routes, doses, parasite and mouse strains, making comparisons challenging. In general, it was not clear why different infection routes, dosages and parasite strains used throughout the manuscript. In addition to describing the logic behind the choices, it would also help the reader if the strains were identified in the text as well as in the figure legend.

3. It would be useful for the authors to define their use of the term "replicating *T. gondii*" (line 133) and if and how it differs from "replicating parasite vacuoles" (line 179). Are vacuoles being counted or tachyzoites being counted?

4. The analysis investigating the distance of the parasites from the nearest capillary in Figure 3 requires some additional clarifying information either in the legend or in the corresponding Methods section. Are the foci being categorized as intravascular/perivascular/parenchymal and the distance for individual parasites within the foci are being measured? The text is unclear. In addition, the authors claim that the extravascular parasites are in close proximity to the capillaries. While the representative images are compelling, the significance of the quantifications being performed is confusing. Why is the comparison between intravascular and extravascular foci relevant (Figure 3i-j)? Perhaps if these data were compared to data from the later D10 time point, the conclusion that the invading parasites are in the vicinity to the capillaries would be easier to substantiate.

5. The data showing brain parasite load in FAK KO mice (Figure 9) was less compelling than the impact of the FAK inhibitor (Figure 8). How do the authors reconcile a lack of impact on parasite numbers (9b), given the increase in numbers of parasite foci (9e) and number of vacuoles/foci (9f) reported? The authors should provide more information about how they quantified the parameters in Figures 9e, f.*Reviewer #3:*

There is a lot to like in this submission as it is a focused on characterizing in detail some of the early events in neuro-invasion of the parasite Toxoplasma. There is some overlap in the early figures with the previous Konradt paper (like vessel size < 10 um, branching, foci of replication, comparison of other tissues and the distinct perivascular organization early after invasion). However, it quickly move past that and attempts to focus on whether the BBB has some important protective effects to neuro-invasion. This is an accepted concept that has really not been tested and the authors also use the increased parasite numbers in lungs and liver compared to brain to promote this idea (while the other explanation is that anything particulate injected iv end up in the lungs and liver because of the blood flow and large numbers of phagocytes and capillary beds).

The authors chose to focus on FAK – which they previously studied in vitro and the most novel aspects of the report are when they start to address the role of this pathway in vivo. Here, there are some concerns. FAK is described as "intercellular junction regulator" but it does much more than that and there is a (somewhat contradictory) literature on its role in Toxoplasm infection – which make this work timely. For example. the PLOS Pathogens 2017 *Toxoplasma gondii* induces FAK-Src-STAT3 signaling to prevents parasite targeting by autophagy – while Loden suggests Toxoplasma targets FAK to disrupt integrin activation and promote hypermotility of infected cells. For the figures 8/9 that address this question the authors need to bolster some of their data sets to make this more compelling (impact on the immune response and recruitment) – rather than leave some obvious gaps that leave a reader unsure whether the data support the conclusions.

The other major gap is that the authors propose on line 609 that testing which route of entry is relevant is beyond the scope of the current submission. Their 2019 publication proposed FAK is important in vitro for transmigration – and disruption of FAK does a lot of things (detailed below). These authors have a real opportunity to test whether altering the BBB affects parasite entry. Providing the can validate the in vivo interventions they are using to target the BBB, the authors should be able to utilize replication deficient parasite strains (to recapitulate the studies of Konradt) and show whether these free parasites in the blood have an improved ability to cross the BBB when FAK is targeted. It's the whole basis for the model and this is the opportunity to test this directly. This reviewer is agnostic to the outcome, but if these parasites do not access the parenchyma more readily in the KO or inhibitor treated mice then the major conclusion (title) would have to change (which is ok). Its also a way of bypassing all the questions about secondary effects of FAK inhibition.

Figure 1 recapitulates the original studies of Konradt that identified infected endothelial cells after iv transfer of parasites and showed preferential localization to small blood vessels. Here the authors also include staining for a5laminin and P-glycoprotein.

Figure 2 – confirms the brain is lower than liver/lungs (again described by Konradt and this could be supplemental) but do a nice job of characterizing location within 3 compartments of the CNS (parenchyma, choroid plexus and meninges).It is unclear from panel E how they identified the choroid plexus.

Figure 3 – is devoted to rigorously establishing that there is a transition from blood, to perivascular to parenchymal localization and that it's the same for different host-parasite combination. The different host-parasite combinations (could be supplemental).

Figure 4 is done with hi and low dose ME49 (moderately virulent) top look at kinetics of immune infiltration. Figure 4C shows increased numbers of CD45+ cells associated with the CNS and this is interpreted as increased inflammation at day 4. Sorry – it was not clear if the brains were perfused for this analysis. Did the authors perform an iv labeling experiment to show that these CD45+ cells are in the brain or just in the blood and are expanded as part of the general response to infection.

Figure 5 uses the virulent RH – and shows changes in BBB permeability that correlates with parasite numbers (more parasites more marker gets across). Basically, RH parasites injected iv for 5 days – these mice are systemically septic. Figure 5F is not significant but the authors conclude that there are "focal permeability elevations in the vicinity of parasite foci.". The data set quite clearly show no significant difference and if they want to focus on those few data points that are >60, then it would be just as important to highlight those few data points that are <25 (which would imply focal sites tighten the BBB). Basically, the authors should consider simplifying their conclusion.

Figure 6 looks at some target genes in MBECs from uninfected mice challenged in vitro with *T. gondii* and conclusion is that there is some upregulation of ICAM. A little hard for me t o know what to do with this data set. Isn't this where they should be looking at FAK dependent integrin expression?

Figure 7B – the authors note there were 5-10 mice /group – but some have only 3 or 4 points. It is also unclear if this is a single experiment or pooled data sets? Just looking at the data sets for the LPS treatment (Figure 7B) – it does look suspiciously like they may have more parasites (hence the question about repeats) while the micrograph in 7C and data in 7f looks like a lot more parasites. This is interesting but whether they used hydro-corticosteroids or daily low dose LPS they see more parasites in the parenchyma. It is also unclear why the control mice in this experiment do not have any parenchymal parasites when they have them in the earlier figures at an earlier time point. Maybe they just need to clarify this. While there is real value to including LPS as a comparator for BBB permeability – daily low dose LPS does a lot of things and I am (honestly) not clear that this figure helps – especially because they have much more specific interventions in the next figures.

Figure 8a – looks like it is underpowered to conclude there is no difference in parasite burden at high dose (a) but there clearly is a difference in all tissues with low dose (b). This suggests a global impact of the inhibitor not an activity restricted to the CNS.

Figure 8 – supplemental the concern here is that the analysis of the IFN-g mRNA looks like it is underpowered and its in the brain not the periphery when the authors indicate that there is a not a lot of local inflammation. If the FAK inhibitor has systemic effects on IFN-g then it would result in more parasites and this assay was used in Supp Figure 7 where it surprisingly showed no impact of hydro-corticosteroids. The authors just need to nail down some better assays for impact on the immune response (serum IFN-g protein?). It is also not clear that more parasites and bigger vacuoles are due to early entry. They have more parasites in the periphery which suggests a defect in ability to control them and this would lead to more early entry and increased growth that is independent of any impact on the BBB. This is really a difficult conclusion to make, with the data sets in hand.

Figure 9 – key experiment – nicely validate FAK deletion and at day 6 post-infection parasite burden is normal across all tissues (differs from effects of inhibitor) and there is no impact on BBB permeability – but the authors find in panels e and f that there are some more parenchymal parasites at day 6. The main conclusion of this submission really revolves around these (similar) findings in Figure 8 and 9 and this reviewer is concerned that there are other explanations. Early in the manuscript they look at immune infiltration but have not seen if that parameter is altered in Figure 8 or 9. Any alteration in immune recruitment/access/function to the CNS is likely to impact parasite control (as they showed in their earlier figures).

---

## [Author Response]

Essential revisions:Detailed comments from reviewers are copied below for your reference. The reviewers were enthusiastic about the potential of the study; however, there was a strong sense that the main hypothesis and title of the study should directly tested as detailed in point 1 below. The reviewers felt that any outcome would be complementary to the present results and should be reported alongside them, either validating the hypothesis or modifying the conclusions of the manuscript.1. Request for additional data. The authors should provide direct evidence for the hypothesis that FAK inhibition or deletion in the vascular endothelium alters parasite translocation by specifically altering barrier function. The reviewers note the various confounding effects of FAK disruption, which extend beyond its role in regulating BBB intercellular junctions. The reviewers were unanimous in requiring experimental evidence using non-replicating strains of Toxoplasma (like CPS) to test this hypothesis since, based on the current model, intravenous injection into WT and BBB FAK KO mice would result in increased translocation of the non-invasive parasites in the latter.

As requested by the reviewers, we have performed experiments with replication-deficient parasites (CPS). First, because the basic invasive features of the CPS line had not been documented, other than ability to invade fibroblast monolayers over an extended incubation time (Fox and Bzik, Nature, 2002), we characterized the gliding motility and the transmigration abilities of this mutant in vitro. Further, CPS has been reported to invade brain endothelium and was very rarely retrieved in non-endothelial /brain parenchymal cells in comparison with wildtype (RH) parasites (Konradt et al., Nat Microbiol, 2016). We undertook a similar approach with the aim of assessing the role of FAK.

The data is provided in the new Figure 9 and Figure 9-Supplements 1, 2 and 3. In summary, the in vitro data shows that the CPS mutant has the ability to transmigrate, albeit at a lower frequency across Caco2 cells (high transcellular electrical resistance (TCER)) and similar to WT across bEND.3 monolayers (moderate-to-high TCER). Interestingly, both transmigration frequencies and gliding path lengths are rapidly reduced upon uracil deprivation. Note also that the overall decrease in gliding is likely underestimated for CPS as only tachyzoites with gliding trails were assessed and an increasing proportion of CPS tachyzoites exhibited no gliding upon uracil deprivation. Over an extended incubation time, CPS invaded cellular monolayers, consistent with reports in fibroblasts (Fox and Bzik, Nature, 2002). Importantly, despite the reduced invasive properties of this mutant, treatment of monolayers with FAK inhibitor consistently led to elevated transmigration frequencies, proportionally similar to effects in WT. This indicates that dysregulation of the tight junctions via FAK (but in absence of monolayer disruption or depolarization) facilitates transmigration for this mutant, as previously shown for WT parasites (Ross et al., cell microbiol, 2019). See Results section (pages 18-19), Figure 9g-j and Figure 9-supplement 2.

In vivo, we noted that very few CPS parasites were retrieved in organs. We therefore opted for a co-infection approach of CPS and WT for more precise comparisons. Similarly, very few invasion events of cells by CPS in different tissues were observed related to WT in peripheral organs and, naturally, even lower numbers in the brain. Thus, despite that dramatically high doses of CPS were inoculated (> 10^8^ freshly egressed tachyzoites i.v., which can be lethal in < 24 h for WT) few parasites are retrieved in brain and in other organs. This made comparisons difficult with WT for total parasite loads and impossible for reliable quantification of translocation events. Specifically, our aim was to normalize data of invasion of endothelial cells by WT and CPS and relate the endothelial invasion numbers to numbers of parasites in the parenchyma. However, with so few invasion events of endothelial cells by CPS, this was not feasible (and inoculation of higher numbers of WT parasites for comparison are rapidly lethal). As an illustrative example and reference: for any strain inoculated, the parasite loads in brain related to peripheral organs are up to 100 times lower (Figure 2). Similarly, the parasite loads in the brain are up to 100 times lower for CPS compared to RH. Jointly, this makes non-replicating CPS parasite loads in brain minimal (that is, up to 100 x 100 = 10.000 times lower than in peripheral organs at day 1 post-inoculation for wt), making in situ quantifications very uncertain.

We compensated with high doses of CPS (> 10^8^) which are lethal and not tolerated for wt. Our assumption, based also on the (surprising!) absence of objective symptoms in mice despite extremely high doses, is that, in vivo, very few CPS parasites manage invasion of cells and that high numbers of parasites are very rapidly cleared (in less than 24 h). Jointly, this underscores further the restrictiveness of the BBB for invasion events. We provide, however, data of pharmacological FAK inhibitor treatment and ECFAK KO mice challenged with CPS. Taking into account the low numbers of retrieved parasites, the data show non-significant differences in total parasite counts. Stainings of brain tissue are also provided with intravascular and extravascular localization of CPS in the brain parenchyma. See Results section (p. 18-19), Figure 9k-o, and Figure 9-supplement 3.

In summary, our data are overall in line with data in Konradt et al., that shows dramatically reduced numbers of CPS-infected/associated/ non-endothelial cells in the brain parenchyma (but not a total absence of CPS-infected cells in brain parenchyma, Konradt-Figure 5i). Our findings ad that a comparable dramatic reduction of intravascular parasites/infection of brain endothelial cells also takes place upon challenge with CPS, related to WT. In our opinion, this precludes direct comparisons with WT parasites for absolute numbers of translocation events. As indicated above, calculations of the ratios infected endothelial cells/infected parenchymal cells for WT and CPS, respectively, were not feasible based on the overall scarce numbers of CPS parasites in brain. Our interpretation is that, in vitro, this difference in invasion is not easily apparent as CPS invades endothelial cells over time (many hours), while invasion of brain endothelium and across brain endothelium in vivo are dramatically compromised. Given the high TCER of the BBB, this result must also be related to the expected overall reduced invasive and transmigration abilities of this mutant, in combination with a sensitivity limitation of the assay due to scarce total invasion events.

We agree with the reviewers that any outcome of the experiments above is complementary to the results. We provide an in vitro characterization of the transmigratory and motility properties of the CPS line with proof-of-concept that transmigration occurs in absence of replication and that FAK inhibition exacerbates transmigration in vitro. In the revised discussion, we reason that, while lysis on infected endothelial cells -and of epithelial cells in other sites, e.g. the gut (Dubey et al., J Euk Microbiol, 1997)- undoubtedly mediates growth across restrictive biological barriers, also other pathways of translocation likely contribute (Matta et al., Nat Rev Microbiol, 2021). Further, because different *T. gondii* genotypes have different invasive phenotypes (Barragan and Sibley, JEM, 2002) and rely differently on transportation by leukocytes (Lambert et al., Infect Immun, 2009) during primary infection, these aspects require further study. We agree that a non-replicating mutant with maintained invasive phenotypes would be a very welcome and extremely useful tool to address these key questions. See discussion (p. 26, lines 708-746)

2. Request for textual modifications or clarification. The itemized reviews list several requests for clarification that can be addressed through changes to the text and the figures. Please respond to all such requests in the resubmission.

Please, find below, point-by-point responses to the queries raised by the reviewers.

Reviewer #1:The authors have generated a detailed and quantitative picture of brain tissue infection following intravenous inoculation with Toxoplasma gondii. The results align well with findings from Konrad et al., (2016) using a similar model. Importantly, the authors examine how entry of parasites into the parenchyma is affected by a variety of perturbations. While many of the perturbations used ultimately affect parasite translocation into the brain, they lack the precision to make specific mechanistic conclusions. Moreover, the authors fail to dissect the contribution of parasite replication within endothelial cells, which would allow them to firmly establish the contribution of the blood-brain barrier in restricting parasite translocation into the brain.1. It seems critical to determine what would be expected from randomly localized vacuoles. E.g. In Figure 3, what would be the average distance of random points within the imaged section from a capillary or a vascular branching point, and what would be the diameter of randomly selected capillaries within a field. Only then can we know whether the observed distribution is unusual.

This is an interesting question on the relative localization of parasite vacuoles and vasculature. In Figure 1, we show that the intravascular parasites preferentially localize to capillaries and to some extent to post-capillary venules. However, these analyzes were not provided for perivascular/parenchymal or randomly located vacuoles. We have now performed additional analyzes on randomly located parasite vacuoles and show the distribution of distances to surrounding vasculature and their diameter within the same field of view. Overall, the localization of parasite vacuoles in relation to vasculature does not follow a normal or random distribution. Rather, the data show an uneven distribution of distances between vacuoles and vasculature, in most cases with a clear vicinity (<40 micrometers) to one capillary (a vessel with luminal diameter < 10 micrometer). The -*putative*- original capillary of infection/translocation is easily identified for intravascular vacuoles and for the vast majority of perivascular vacuoles (with a clear difference to average distances to any other vessel). However, note that there is more spread of the data among the parenchymal vacuoles (for example, 2 different capillaries can be equidistant to a parenchymal vacuole), consistent with the purpose of the dissemination process(es) within the parenchymal tissue (Figure 3f and Figure 3-supplement 2). However, for most parenchymal vacuoles, the vicinity to other (intravascular, perivascular) vacuoles provides the indication of the -putative-original capillary from which translocation was effectuated (Figure 3e).

The new compiled data has been added to the new Figure 3f and representative detailed analyses are provided in new Figure 3- Supplement 2, Results (p. 7, l. 175-). We have added a comment in the discussion (p. 22-23, l. 604-8).

2. Can the authors firmly conclude that at low parasite levels there is no permeability? The authors should consider whether single translocation events would provide sufficient signal to detect vascular leakage.

This is a highly relevant question that we´ve given quite some thought past years and we fully agree with the reviewer (absence of evidence is not evidence of absence). The complexity lays in how “permeability” is defined. While a subtle vascular leakage below the detection threshold of the marker Evans Blue cannot be excluded, our data advocate against (1) a generalized BBB dysfunction with enhanced BBB permeability (as in LPS control treatment) or (2) an irreversible BBB damage with hemorrhage (as seen in malaria or meningococcal petechial bleedings). However, because we measured (3) sparse and focalized permeability elevations in the vicinity of parasite foci, this may indicate permeability variations over time and/or reversible permeability elevations, in line with a transient BBB dysregulation.

Thus, it cannot be ruled out that one single parasite translocation event (for example, following egress from endothelium or by direct transmigration) locally and reversibly destabilizes the BBB in vivo below the detection threshold of Evans Blue. In fact, although in vivo direct evidence is hard to obtain, we consider this as a highly likely possibility based on in vitro data (Ross et al., Cell Micro, 2019; Ross et al., CMLS, 2021). in vitro, extracellular tachyzoites or parasitized dendritic cells consistently transmigrated without a measurable irreversible disruption (defined by fluorescent low-molecular weight tracer and transcellular electrical resistance (TCER)) of the barrier. Importantly, a transient dysregulation of polarized monolayers was picked up by a minor but consistent decrease in TCER but without measurable leakage of the fluorescent tracer (Ross et al., Cell Micro, 2019). This dose-dependent effect was more subtle for *T. gondii* challenge compared to LPS challenge, that readily impacted polarization and permeability. Thus, our in vivo observations here are well in line with in vitro data. In summary, we cannot firmly exclude absence of permeability and the data jointly favor a situation encompassing dysregulation (but not irreversible disruption) of the BBB.

We have further clarified this important aspect and our reasoning in the discussion (p 24-25, l. 659-80).

3. The authors need to exclude replication within endothelial cells to firmly establish that the BBB restricts parasite translocation. This could be accomplished by inoculating parasites incable of replication in mice. Without this evidence, it is formally possible that FAK inhibition/knockout or LPS are simply affecting parasite replication within endothelial cells.

We thank the reviewer for this constructive suggestion. We have addressed our results with replication-deficient parasites at the beginning of this rebuttal and refer to that response to avoid redundance. in vitro, FAK inhibition or knockdown of endothelial FAK by shRNA did not significantly impact parasite replication (WT) within endothelial cells (Ross et al., Cell Microbiol, 2019).

4. Related to point 3, the number of parasite foci trends up in both perturbations of FAK (inhibition and knockout), within the intravascular compartment. Are investigators truly capable of distinguishing between perivascular and intravascular infection, given the lack of a specific marker for endothelial cells? The concern is that in some cases leakage of the endothelium will land parasites in perivascular sites that would look microscopically very similar to vascular sites; this effect could explain the surge in intravascular foci for perturbations that should alter permeability.

The reviewer brings up an interesting thought.

The differences between intravascular and perivascular localizations can be best appreciated in the provided 3D reconstructions in the videos and are also illustrated in Figure 3a. For example, S4 video shows examples of intravascular, perivascular (and parenchymal) vacuoles in the same field of view. Note also that the perivascular localization can occur in absence of intravascular vacuoles (S10 video).

As the reviewer points out, we cannot exclude that, in some rare cases, ambiguous localization might happen but looking at the totality of the data we feel confident that what we define as “perivascular” (>10 μm from the EB signal of the vascular lumen) or “intravascular” (within the luminal diameter of the vessel and 10 from the EB signal of the vascular lumen) can be discerned both by 2-photon microscopy in fresh tissue and by confocal in fixed tissue.

Note that our description in the paper is related to the luminal signal by EB and that no claim is made is relation to the precise cellular localization. However, it would seem that while intravascular parasites are tightly associated with the endothelial markers alpha4/beta1 laminin and P-glycoprotein, perivascular parasites localize more with GFAP and Iba1 marker (Figure 1, Figure 3). The perivascular localization is, we agree, more difficult to define but also serves the purpose of describing this intermediate localization of parasites in relation to the vascular lumen which is clearly distinct from the more disperse parenchymal localization. We think that this -descriptive categorization- is possibly more faithful to the observations, rather than classifying parasites as strictly parenchymal and strictly intravascular. And, we agree, future investigations, must pin down the precise localization of these parasites and possible host cells (pericytes? Astrocyte end-feet? Microglia? Perivascular macrophages?).

The reviewer is also correct that upon counting hundreds of foci, there are examples that are more difficult to determine than others. Our opinion is that invasion is a continuous process, while parasite replication takes place in the different locations and the focus grows successively. The different localizations are not mutually exclusive and co-exist (S4 video).

For FAK inhibition, the reviewer correctly points out that the numbers and size of intravascular foci augment compared to the perivascular and parenchymal foci for the high dose/short time (day 3) challenge (Figure 8d, f). However, for the low dose/long time (day 8) challenge the relative increase in vacuole numbers is more accentuated for the parenchymal localization (Figure 8e) with a general size increase of the foci (Figure 8g). A comparable increase in vascular permeability is observed in both conditions (Figure 8h, i). In Figure 9, the increase in parenchymal localization in proportionally higher compared to intravasc/perivasc. Thus, FAK inhibition or KO does not seem to automatically lead to a relative higher increase in intravascular parasites in relation to the two other localizations (perivascular and parenchymal) for all conditions. However, we agree that pharmacological FAK inhibition might have other effects that impact parasitemia or even parasite survival in endothelium. These aspects are outlined in the discussion (p. 25, l.690-).

In case of hemorrhage, the idea that “perivascular sites would look very similar to vascular sites due to enhanced permeability” is appealing but would likely entail a strong focal concentration of the EB marker with parenchymal diffusion (“bleeding”) and increased “diameter” of the affected area (would presumably look as if a larger irregular blood vessel was infected?). We did not observe this type of event. We carefully assessed this because it was, in our opinion, crucial to define if hemorrhagic components take place. Alternatively, there could be a milder form of permeability elevation without hemorrhage. We favor this alternative. In Figure 5-Supplement 1, the intensity of the fluorescent marker (EB) between lumen and non-luminal varies significantly. Hypothetically, this situation would be picked up as an intensity elevation with increased “diameter” of the blood vessel, but we did not make that type of observation.

We also carefully assessed foci with combined intravascular, perivascular and parenchymal (see, red-colored foci in Figure 3e) and did not observe dramatically enhanced signal of blood/permeability tracer in these foci, compared with other foci. Thus, although it cannot be fully excluded, we think that the situation proposed by the reviewer cannot be a common or predominating situation.

We have clarified these aspects in the discussion (p 22-23, l. 690-, p 24-25, l. 659-)

Reviewer #2:1. There is some concern that i.v. injection of large numbers of free tachyzoites might not accurately recapitulate events during normal infection. In addition the majority of experiments used the highly virulent non-cyst forming strain of the parasite (RH), which may use a different route of entry compared to less virulent, cyst forming strains. The authors should acknowledge and address these caveats in the text.

The reviewer brings up several important aspects related to the validity of the model and variations among strains. For this reason, we used both ip and iv routes and type I and type II strains. In fact, a higher number of figures/suppl. Figurescontain ME49 data (13) compared to RH data (9).

We observed a similar distribution of parasites for the two routes and the two parasite strains. Additionally, permeability analyses were consistent. Jointly, this argues for consistency between type I vs type II and between iv vs ip routes. However, we agree that there are limitations: (1) To be able to consistently obtain quantifiable data, high numbers of parasites had to be injected. This has the drawback of reducing mouse survival/shortening days of infection. (2) For longer mouse survival (day 5-10), lower doses of type II strain were given.

In the manuscript, we have clarified further the rationale for using different mouse strains and parasite lines, with a reference (Results, p. 7, l. 176-88). We acknowledge the limitations of the iv/ip infection model and its possible bias for passage in the discussion (p. 22, l. 594-604).

2. The conclusion that parasite entry (days 3-5) precedes BBB permeability, leukocyte entry and inflammation (day 10) is drawn from comparison of data from Figures 4, 5 and 6. However, the experiments in these 3 figures uses a variety of different infection routes, doses, parasite and mouse strains, making comparisons challenging. In general, it was not clear why different infection routes, dosages and parasite strains used throughout the manuscript. In addition to describing the logic behind the choices, it would also help the reader if the strains were identified in the text as well as in the figure legend.

We agree that this required further clarification. It has to be kept in mind that localization of *T. gondii* in brain vasculature and brain parenchyma are relatively rare events (related to number of parasites in circulation or parasite loads in the parenchyma of other peripheral organs, see for example Figure 2). Consistent with this, relatively few cysts are formed in the brain in natural infections. Within this biological setting (rare events), technical considerations had to be taken into account in order to repeatedly obtain reproducible and quantifiable data in our study and similarly in other studies with similar approach. Specifically, the caveats of experimental systems (for example unnaturally high doses and unnatural infection route) have to be weighed against the benefit of acquisition of quantifiable and reproducible data. If this type of data could reasonably be obtained by natural oral infections, then that would definitively be the way to go.

We agree that determining the exact timepoint when *T. gondii* crosses the barrier is not an easy task. In this paper, we attempt to document the earliest possible timepoints of CNS penetration, for each condition and mouse/parasite strain. This does not exclude passage at later timepoints. In fact, it is reasonable to consider passage/translocation as a continuous process during primary infection, until parasitemia ceases. However, we admit that we were surprised by the rapid presence of parasites in the parenchyma.

The reason for using different doses, routes and strains have been brought up above and we have clarified this in the text (Results, Material and Methods). For each experiment, the strain, route and dose used are indicated in the figure legends, relevant for the general conclusions.

We have also clarified these aspects in the revised Discussion (p 22-23, l. 594-, p 24, l. 643-46, p. 25, l. 690-).

3. It would be useful for the authors to define their use of the term "replicating T. gondii" (line 133) and if and how it differs from "replicating parasite vacuoles" (line 179). Are vacuoles being counted or tachyzoites being counted?

Vacuoles with replicating *T. gondii* tachyzoites (2 or more per vacuole) were counted.

This has been clarified (results, p. 5, line 129, andp. 7, l. 179) and precision of the definition of the counting of parasite foci/parasite vacuoles/single tachyzoites has been added to Methods (p. 29).

“….A parasite focus was defined as a group of vacuoles with replicating T. gondii (GFP^+^). Unless differently stated, individual vacuoles with replicating T. gondii tachyzoites (2 or more tachyzoites per vacuole) were counted. The size of the foci was determined by counting number of vacuoles and categorized as follows: <10, 11-50 or >50 vacuoles/focus. Single tachyzoites were also counted in Figure 3e and 3h-k….”

4. The analysis investigating the distance of the parasites from the nearest capillary in Figure 3 requires some additional clarifying information either in the legend or in the corresponding Methods section. Are the foci being categorized as intravascular/perivascular/parenchymal and the distance for individual parasites within the foci are being measured? The text is unclear. In addition, the authors claim that the extravascular parasites are in close proximity to the capillaries. While the representative images are compelling, the significance of the quantifications being performed is confusing. Why is the comparison between intravascular and extravascular foci relevant (Figure 3i-j)? Perhaps if these data were compared to data from the later D10 time point, the conclusion that the invading parasites are in the vicinity to the capillaries would be easier to substantiate.

We agree that these aspects needed additional precision for clarity. We have clarified the definitions of focus/vacuole/single tachyzoite (see above). The reason for using “*T. gondii* tachyzoites (GFP+)” in the figure legend for micrographs is that GFP expression by tachyzoites are used to identify vacuoles with replicating tachyzoites and not a vacuole marker *per se* (we wanted to avoid this possible confusion). We have provided precision to Methods (p. 29).

For Figure 3i-j. We think that, overall, these analyzes are relevant because they show that the blood vessel diameter, distance ranges, frequencies and distribution of parasites in live (fresh, unfixed) tissue analyzed by 2-photon microscopy are comparable to those of fixated tissue (processing and fixation can sometimes generate artefacts with distortion of distances and localization), thereby validating and reinforcing the data using fixed tissue in Figure 3e, f. We have further clarified the rationale (p. 7, l. 187-).

The new Figure 3f provides additional data on the distances of randomly located foci to the vasculature. We think this will clarify further to the reader that the localization of *T. gondii* in relation to a capillary is not a random event (see also response to question 1 by Reviewer 1).

5. The data showing brain parasite load in FAK KO mice (Figure 9) was less compelling than the impact of the FAK inhibitor (Figure 8). How do the authors reconcile a lack of impact on parasite numbers (9b), given the increase in numbers of parasite foci (9e) and number of vacuoles/foci (9f) reported? The authors should provide more information about how they quantified the parameters in Figures 9e, f.

The experiments required a number of considerations. First, FAK inhibitor studies allowed the use of CD1 outbred mice (allowing assessment at day 8 post-infection), significantly more resistant to *T. gondii* compared to B6 mice (allowing assessment max. at day 6 post-infection based on criteria of our animal ethics permit). Thus, a direct comparison of the two treatments cannot be directly done because mouse strain, doses and days differ. We have indicated the conditions for each experiment in figures legends.

For the differences between parasites numbers by plaquing counts (9b), we think the two methods measure partly different things and are complementary. Plaquing assays quantify total parasite numbers from dilutions of homogenized tissues, which may be less precise compared to counting numbers of in situ parasite foci (9e) and vacuoles/foci (9f). Plaquing assays have the advantage though that they are representative of the totality of the tissue, while counts of parasite foci provide insights locally in the parenchyma. We have added precision to the discussion and specifically discuss differences between pharmacological treatment and knockout (p. 25, l. 690-706).

For the quantitative parameters and definitions, more precision is now provided under Methods (p. 29).

Reviewer #3:The authors chose to focus on FAK – which they previously studied in vitro and the most novel aspects of the report are when they start to address the role of this pathway in vivo. Here, there are some concerns. FAK is described as "intercellular junction regulator" but it does much more than that and there is a (somewhat contradictory) literature on its role in Toxoplasm infection – which make this work timely. For example. the PLOS Pathogens 2017 Toxoplasma gondii induces FAK-Src-STAT3 signaling to prevents parasite targeting by autophagy – while Loden suggests Toxoplasma targets FAK to disrupt integrin activation and promote hypermotility of infected cells. For the figures 8/9 that address this question the authors need to bolster some of their data sets to make this more compelling (impact on the immune response and recruitment) – rather than leave some obvious gaps that leave a reader unsure whether the data support the conclusions.The other major gap is that the authors propose on line 609 that testing which route of entry is relevant is beyond the scope of the current submission. Their 2019 publication proposed FAK is important in vitro for transmigration – and disruption of FAK does a lot of things (detailed below). These authors have a real opportunity to test whether altering the BBB affects parasite entry. Providing the can validate the in vivo interventions they are using to target the BBB, the authors should be able to utilize replication deficient parasite strains (to recapitulate the studies of Konradt) and show whether these free parasites in the blood have an improved ability to cross the BBB when FAK is targeted. It's the whole basis for the model and this is the opportunity to test this directly. This reviewer is agnostic to the outcome, but if these parasites do not access the parenchyma more readily in the KO or inhibitor treated mice then the major conclusion (title) would have to change (which is ok). Its also a way of bypassing all the questions about secondary effects of FAK inhibition.

We thank the reviewer for the insightful and constructive comments. We agree with the point of view that FAK is a molecule implicated in many biological processes and have clarified these aspects further in the revised discussion with additional references (p. 25, l. 690-706). Further, we have characterized the transmigration of replication-deficient parasites (CPS) in vitro and in vivo.

Figure 1 recapitulates the original studies of Konradt that identified infected endothelial cells after iv transfer of parasites and showed preferential localization to small blood vessels. Here the authors also include staining for a5laminin and P-glycoprotein.

Yes, the data in Figure 1 (3 days post inoculation (dpi), RH strain, B6 mice) is consistent with the data in Konradt et al., (7-9 dpi, RH strain, B6 mice, Konradt-Figure 3f) showing that *T. gondii* preferentially localizes to vessels with a lumen < 10 micrometers. We felt it was crucial to firmly establish this preferential localization at the earliest possible timepoint – permitting reliable quantifications- because parasites infecting endothelium by day 7-9 could either have replicated locally earlier days or have replicated in other peripheral organs and later been transported with the cerebral circulation. Later in the paper, we also confirm this result using outbred mice (CD1) and another parasite strain (ME49, type II) (Figure 3f). This needed more precision to the reader and we have clarified these aspects in the text (p. 23, l. 609-11).

Figure 2 – confirms the brain is lower than liver/lungs (again described by Konradt and this could be supplemental) but do a nice job of characterizing location within 3 compartments of the CNS (parenchyma, choroid plexus and meninges).It is unclear from panel E how they identified the choroid plexus.

For parasite loads, we reason as above because it was important to determine if the dramatic differences in parasite loads between brain and other peripheral organs with parenchyma (≈10-100 fold depending on timepoint and dose) occur very early (even after iv injection with rapid systemic spread of parasites) and depend on mouse strain, or if they would depend primarily on differences in parasite replication and dissemination (or combinations). In a sense, because the differences in parasite loads are present from day 1 (before significant parasite expansion takes place), similarly in both mouse strains, we think these data indirectly illustrate the restrictiveness of the BBB compared to other endothelial barriers in peripheral organs and motivate a further exploration of the BBB. This may indicate that the brain vasculature is less permissive to adhesion or invasive events by *T. gondii*. Precision has been added to the text (discussion, p. 22, l. 584-).

The choroid plexus was identified by its anatomical localization and its unique morphological structure upon staining and perfusion of Evans Blue dye. A Suppl. figure has been added to the manuscript (Figure 2—figure supplement 1).

Figure 3 – is devoted to rigorously establishing that there is a transition from blood, to perivascular to parenchymal localization and that it's the same for different host-parasite combination. The different host-parasite combinations (could be supplemental).

Figure 3 is complemented with a supplementary Figure that contains the different host-parasite combinations (Figure 3- Supplement 1c-f).

Figure 4 is done with hi and low dose ME49 (moderately virulent) top look at kinetics of immune infiltration. Figure 4C shows increased numbers of CD45+ cells associated with the CNS and this is interpreted as increased inflammation at day 4. Sorry – it was not clear if the brains were perfused for this analysis. Did the authors perform an iv labeling experiment to show that these CD45+ cells are in the brain or just in the blood and are expanded as part of the general response to infection.

The purpose of these experiments was to determine the total leukocyte infiltration, specifically at the early timepoints and how leukocyte numbers (intravascular and extravascular) relate to the early passage of *T. gondii* (localization in the brain parenchyma). Brains were therefore not perfused and localization of leukocytes in the brain parenchyma is instead shown by immunofluorescence stainings (Figure 4-Suppl 1). The reviewer is correct that part of the abundant leukocyte infiltration could emanate from expansion (Schneider et al., PNAS, 2019) however the main point here was that some parasites enter the CNS before a significant leukocyte infiltration is observed. We wanted to provide the reader with the information that measurable passage occurs in absence of an important infiltrative process and before day 10. We have clarified this in the text (p. 24, l. 541-47)

Figure 5 uses the virulent RH – and shows changes in BBB permeability that correlates with parasite numbers (more parasites more marker gets across). Basically, RH parasites injected iv for 5 days – these mice are systemically septic. Figure 5F is not significant but the authors conclude that there are "focal permeability elevations in the vicinity of parasite foci.". The data set quite clearly show no significant difference and if they want to focus on those few data points that are >60, then it would be just as important to highlight those few data points that are <25 (which would imply focal sites tighten the BBB). Basically, the authors should consider simplifying their conclusion.

We write “Overall, non-significant differences in permeability were observed when comparing brain sections containing tachyzoites (GFP^+^) and sections with absence of tachyzoites (GFP^-^) at days 3 or 10 (Figure 5e, f). However, a subset of GFP^+^ vessels exhibited elevated EB signal, (Figure 5f).

The reason for highlighting to the reader that a few foci exhibited non-significant but higher permeability is that this could -hypothetically- be related to passage. It is hard to speculate what the subset with reduced permeability represents (countereffect?). Possibly a larger material (n > > 234 vessels analyzed here) could provide a more definitive answer. We have moved the more speculative part under results (above) to the discussion (p. 24, l. 659-80).

Figure 6 looks at some target genes in MBECs from uninfected mice challenged in vitro with T. gondii and conclusion is that there is some upregulation of ICAM. A little hard for me t o know what to do with this data set. Isn't this where they should be looking at FAK dependent integrin expression?

We agree and have expanded on this. At the time of our submission, another manuscript where we study the roles of integrins and CAMs in vitro was under revision and is now published. By citing these data, we now better link these aspects to possible mechanisms of passage in infected leukocytes with reference to (Ross et al., CMLS, 2021) and also passage/invasion of endothelium and epithelium by extracellular tachyzoites (Barragan et al., CMI, 2005). The putative connection to FAK dysregulation awaits further investigation. Ongoing work indicates a role for CAMs in this process (Ross et al., in preparation). We have provided precision to these aspects in the text and additional references have been added (p. 23, l. 616-23).

Figure 7B – the authors note there were 5-10 mice /group – but some have only 3 or 4 points. It is also unclear if this is a single experiment or pooled data sets? Just looking at the data sets for the LPS treatment (Figure 7B) – it does look suspiciously like they may have more parasites (hence the question about repeats) while the micrograph in 7C and data in 7f looks like a lot more parasites. This is interesting but whether they used hydro-corticosteroids or daily low dose LPS they see more parasites in the parenchyma. It is also unclear why the control mice in this experiment do not have any parenchymal parasites when they have them in the earlier figures at an earlier time point. Maybe they just need to clarify this. While there is real value to including LPS as a comparator for BBB permeability – daily low dose LPS does a lot of things and I am (honestly) not clear that this figure helps – especially because they have much more specific interventions in the next figures.

This point is well taken and we agree that the effects of anti- and pro-inflammatory treatments are probably as complex as the infection processes themselves. We included these treatments because they impact/modulate the infection process in different ways. Yet, as pointed out by the reviewer, both treatments result in elevated parasite numbers. We do not draw any major conclusion from this, other than pro-inflammatory LPS treatment – elevating BBB permeability – impacts parasite numbers and so does anti-inflammatory treatment, likely by dampening immune responses. The total parasite loads are provided as a guideline to the reader for comparisons. We have toned down the argumentation in the discussion and also provided additional nuance (p. 24, l. 649-56)

Related to parasites in parenchyma: When possible, outbred CD1 mice are used (this figure 7 for example) because of their superior resistance to T. gondii compared with B6 mice (figure 3a, b for example). Further, dose and inoculation route naturally impacted parasite loads and mouse survival time. Thus, the conditions for different figures are not necessarily directly comparable and address different aspects.

Specifically in Figure 7, CD1 mice are inoculated with low dose i.p.. Parenchymal parasites can be quantified by day 3 in B6 mice inoculated with high dose i.v. (Figure 3b), however low dose i.p. does not allow reliable quantification by IF by day 6 and therefore only day 10 is shown (Figure 3m). These adjustments (strain/dose/administration route) were done to assure survival of the mice and in order to follow strict guidelines for minimization of severe symptoms, following ethical permit protocols. The rationale for use of different mouse strains, parasite strains and administration routes has been further clarified (results p. 7) and each figure legend indicates mouse strain, dose and administration route.

Related to replicates: The reason for the fewer points in some graphics were technical (specifically for these few points, that overgrowth in plaquing assays due to insufficient dilutions did not allow quantification for these specific data points). These experiments have been repeated and additional datapoints added. The compiled results reinforce the trend previously observed and show significant differences between the conditions (new Figure 7b).

Figure 8a – looks like it is underpowered to conclude there is no difference in parasite burden at high dose (a) but there clearly is a difference in all tissues with low dose (b). This suggests a global impact of the inhibitor not an activity restricted to the CNS.

This is very possible or, also that the relative inhibition for high dose (3 days, high dose) is inferior to the cumulative impact that FAK inhibition has over time (8 days, low dose). Regardless, we do not make major conclusions on this, total parasite loads are given as reference for loads in the brain and we think it does not change the overall conclusions of the paper. We have expanded on the possible “side effects” of pharmacological FAK inhibition in the revised discussion and advantages/disadvantages compared with conditional knockout (p. 25, l. 690-)

Figure 8 – supplemental the concern here is that the analysis of the IFN-g mRNA looks like it is underpowered and its in the brain not the periphery when the authors indicate that there is a not a lot of local inflammation. If the FAK inhibitor has systemic effects on IFN-g then it would result in more parasites and this assay was used in Supp Figure 7 where it surprisingly showed no impact of hydro-corticosteroids. The authors just need to nail down some better assays for impact on the immune response (serum IFN-g protein?). It is also not clear that more parasites and bigger vacuoles are due to early entry. They have more parasites in the periphery which suggests a defect in ability to control them and this would lead to more early entry and increased growth that is independent of any impact on the BBB. This is really a difficult conclusion to make, with the data sets in hand.

We agree and have clarified this figure to better show the variability between individual mice. As discussed above, the effects of treatments are likely multiple and these inflammatory parameters are provided as guidance that responses are not out of range (very high or very low) but rather in a comparable range. This does not exclude local or systemic differences that would require detailed analysis and cannot be made with the data sets in hand.

We have modified the text under results (p. 16, l. 422, Figure 8-supplement 1b, c) and bring up the possible impact of FAK inhibition on immune responses in the discussion (p. 25, l. 690-)

Figure 9 – key experiment – nicely validate FAK and at day 6 post-infection parasite burden is normal across all tissues (differs from effects of inhibitor) and there is no impact on BBB permeability – but the authors find in panels e and f that there are some more parenchymal parasites at day 6. The main conclusion of this submission really revolves around these (similar) findings in Figure 8 and 9 and this reviewer is concerned that there are other explanations. Early in the manuscript they look at immune infiltration but have not seen if that parameter is altered in Figure 8 or 9. Any alteration in immune recruitment/access/function to the CNS is likely to impact parasite control (as they showed in their earlier figures).

The reviewer brings up an important point. In Figure 4 we show that leukocyte infiltration days 1-5 is subtle and becomes accentuated by day 10. Thus, by day 6 it is still not prominent but could be affected by endothelial FAK deletion.

We have performed an additional validation testing expression of FAK in bone-marrow derived cells from mice. By western blotting, we show that while FAK is reduced in endothelial cells, FAK expression is maintained in bone marrow-derived cells (results, p. 18, l. 468-, and Figure 9-supplement 1b, c).

We address the caveats of each system utilized in Figure 8 and 9 in the revised discussion (p. 25, l. 690-)